



# Potential Artifacts in Conservation Laws and Invariants Inferred from Sequential State Estimation

Carl Wunsch[1], Sarah Williamson[2], and Patrick Heimbach[2, 3, 4]

[1]Department of Earth and Planetary Science, Harvard University, Cambridge, MA, USA
[2]Oden Institute for Computational Engineering and Sciences, University of Texas at Austin, Austin, TX, USA
[3]Jackson School of Geosciences, University of Texas at Austin, Austin, TX, USA
[4]Institute for Geophysics, University of Texas at Austin, Austin, TX, USA

**Correspondence:** Sarah Williamson, swilliamson@utexas.edu

**Abstract.** In sequential estimation methods often used in oceanic and general climate calculations of the state and of forecasts, observations act mathematically and statistically as forcings. For purposes of calculating changes in important functions of state variables such as total mass and energy, or volumetric current transports, results are sensitive to mis-representation of a large variety of parameters, including initial conditions, prior uncertainty covariances, and systematic and random errors in
observations. Here toy models of a mass-spring oscillator and of a barotropic Rossby-wave equation are used to demonstrate many of the issues. Results from Kalman-filter estimates, and those from finite interval smoothing are analyzed. In the filter (and prediction) problem, entry of data leads to violation of conservation and other invariant rules. A finite interval smoothing method restores the conservation rules, but uncertainties in all such estimation results remain. Convincing trend and other time-dependent determinations in "reanalysis"-like estimates require a full understanding of both models and observations.

## 1 Introduction

Intense scientific and practical interest exists in understanding the time-dependent behavior in the past and future of elements of the climate systems of a full reanalysis computation, some simple examples of the known difficulties with sequential analysis methods could be useful. Expert practitioners of the methodology, particularly on the atmospheric side (e.g. Dee (2005), Cohn (2010), Janjić et al. (2014), and Gelaro et al. (2017)) clearly understand the pitfalls of the methodologies, but many of these
discussions are couched in the mathematical language of continuous space-time (requiring the full apparatus of functional analysis) and/or the specialized language of atmospheric sciences. Somewhat controversial, contradictory, results in the public domain (e.g. Hu et al. (2020) or Boers (2021)) suggest that, given the technical complexities of a full reanalysis computation, some simple examples of the known difficulties with sequential analysis methods could be helpful. For scientists interested in the results, but not fully familiar with the machinery being used, it is useful to have a more schematic, simplified set of examples
so that the numerous assumptions underlying reanalyses and related calculations can be fully understood. Dee (2005) is close in spirit to what is attempted here. As in geophysical fluid dynamics methods, two "toy models" are used to gain insight into issues applying to far more realistic systems.



Discussion of even the simplified systems considered below requires much notation. Although the Appendix writes out the fuller notation and its applications, the basic terminology used is defined more compactly here. Best estimates of past, present, and future invoke knowledge of both observations and models and both involve physical-dynamical, chemical, and biological elements.

Fundamental to understanding many physical systems is analysis of long-term changes in quantities that are subject to various conservation rules (e.g. energy, enstrophy, total mass, and mean concentrations). Conservation rules in physical systems imply that any changes in the quantity are specifically attributable to identifiable sources/sinks/dissipation in the interior and in the boundary conditions. Absent that identifiability in e.g., mass or energy conservation, claims to physical understanding must be viewed with suspicion. In climate science particularly, violation undermines the ability to determine system trends over months, decades, and longer.

Two major reservoirs of understanding of systems such as those governing the ocean or climate overall lie with observations of the system, and with the equations (e.g., Navier-Stokes) believed applicable. Appropriate combination of the information from both reservoirs generally leads to improvement over estimates made from either alone, but should never degrade them. Conventional methods for combining data with models fall into the general category of control theory, in both mathematical and engineering forms, although full understanding is made difficult in practice by the need to combine major sub-elements of different disciplines, including statistics of several types, computer science, numerical approximations, oceanography, meteorology, climate, dynamical systems theory, and the observational characteristics of very diverse instrument types and distributions. Within the control theory context, distinct goals include "filtering" (what is the present system state?), "prediction" (what is the best estimate of the future state?), and "interval smoothing" (what was the time history over some finite past interval?) and their corresponding uncertainties.

In oceanography, and climate physics and chemistry more generally, a central tool has become what meteorologists call a "reanalysis,"—a time-sequential estimation method based ultimately on long experience with numerical weather prediction. Particular attention is called, however, to Bengtsson et al. (2004) who showed the impacts of observational system shifts on apparent climate change outcomes arising in some sequential methods. A number of subsequent papers (see, for example, Bromwich and Fogt (2004), Bengtsson et al. (2007), Carton and Giese (2008), and Thorne and Vose (2010)) have noticed difficulties in using reanalyses for long-term climate properties sometimes ending with advice—such as "minimize the errors" (see Wunsch (2020) for one global discussion).

For some purposes, e.g., short-term weather or other prediction, system failure to conserve mass or energy or enstrophy may be of no concern—as the time-scale of emergence for detectable consequences of that failure can greatly exceed the forecast time. In contrast, for reconstruction of past states, those consequences can destroy any hope of physical interpretation. In long-duration forecasts with rigorous models, which by definition contain no observational data at all, conservation laws and other invariants of the model are likely to be preserved. Tests however, of model elements and in particular of accumulating errors, are not then possible until adequate data do appear.

We introduce notation essential for the methods used throughout the manuscript in Sect. 2. Experiments that examine the impact of data on reconstruction of invariants in a mass-spring oscillator system are discussed in section 3. This includes the





impact of data density and sparsity on reconstructions of energy, position, and velocity, and ends with a discussion of the
structure of the covariance matrix. Section 4 covers the Rossby wave equation and examines a simplified dynamical system
resembling a forced Rossby wave solution. Here a combination of the Kalman Filter and the Rauch-Tung-Striebel smoother
is used to reconstruct a pseudo-energy as well as the time-independent transport of a western boundary. Results are discussed
in Sect. 5. Discussion of even the simplified systems considered below requires much notation. Although the Appendix writes
out the fuller notation and its applications, the basic terminology used is defined in the following.

## 2 Notation and Some Generic Inferences

All variables, independent and dependent, are assumed to be discrete. Notation is similar to that in Wunsch (2006). Throughout
the manuscript lower case bold variables are vectors and upper case bold variables are matrices.

Let $\mathbf{x}(t)$ be a state vector for time $0 \le t \le t_f = N_t \Delta t$. A state vector is one that completely describes a physical system
evolving according to a model rule (in this case, linear),

$$\mathbf{x}(t + \Delta t) = \mathbf{A}(t)\mathbf{x}(t) + \mathbf{B}(t)\mathbf{q}(t),\tag{1}$$

where $\Delta t$ is the constant time-step. $\mathbf{A}(t)$ is a square "state-transition matrix" , and $\mathbf{B}(t)\mathbf{q}(t)$ is any imposed external forcing,
including boundary conditions, with $\mathbf{B}(t)$ a matrix distributing disturbances $\mathbf{q}(t)$ appropriately over the state vector. Generally
speaking, knowledge of $\mathbf{x}(t)$ is sought by combining Eq. (1) with a set of linear observations,

$$\mathbf{y}(t) = \mathbf{E}(t)\mathbf{x}(t) + \mathbf{n}(t)\tag{2}$$

Here $\mathbf{E}(t)$ is another known matrix, which typically vanishes for most values $t$, and represents how the observations measure
elements of $\mathbf{x}(t)$. The variable $\mathbf{n}(t)$ is the inevitable noise in any real observation and for which some basic statistics will
be assumed. Depending upon the nature of $\mathbf{E}(t)$, Eq. (2) can be solved by itself for an estimate of $\mathbf{x}(t)$. (As part of the
linearization assumption, neither $\mathbf{E}(t)$ nor $\mathbf{n}(t)$ depends upon the state vector.)

Estimates of the (unknown) true variables $\mathbf{x}(t)$ and $\mathbf{q}(t)$ are written with tildes, $\tilde{\mathbf{x}}(t)$, $\tilde{\mathbf{q}}(t)$, $\tilde{\mathbf{x}}(t, -)$, $\tilde{\mathbf{x}}(t, +)$, etc. As bor-
rowed from control theory convention, the minus sign denotes a prediction of $\mathbf{x}(t)$ *not* using any data at time $t$, but possibly
using data from the past. If no data at $t$ are then used, $\tilde{\mathbf{x}}(t) = \tilde{\mathbf{x}}(t, -)$. The plus sign indicates an estimate at time $t$ where data
*future* to time $t$ have also been employed. In what follows, the "prediction" model is always of the form Eq. (1), but usually
with deviations in $\mathbf{x}(0)$, and in $\mathbf{q}(t)$, which must be accounted for.

In any estimation procedure, knowledge of the initial and resulting uncertainties is required. For these linear problems, a
bracket is used to denote expected value, e.g. the variance matrix of any variable $\boldsymbol{\xi}(t)$ is denoted,

$$\mathbf{P}_\xi(t) = \left\langle \left(\tilde{\boldsymbol{\xi}}(t) - \mathbf{a}\right)\left(\tilde{\boldsymbol{\xi}}(t) - \mathbf{a}\right)^T \right\rangle\tag{3}$$

and where $\mathbf{a}$ is usually the true value of $\boldsymbol{\xi}(t)$, or some averaged value. (When $\xi$ is omitted in the subscript, $\mathbf{P}$ refers to $\tilde{\mathbf{x}}$.)

Together, Eqs. (1), (2) are a set of linear simultaneous equations for $\mathbf{x}(t)$ and possibly $\mathbf{q}(t)$ and which, irrespective of
whether over- or under-determined, can be solved by the standard inverse methods from linear algebra. For systems too large





for such a calculation and/or ones in which data continue to arrive (e.g., for weather) following a previous calculation, one

moves to using sequential methods in time.

Suppose that, starting from $t = 0$, a forecast is made using only the model Eq. (1) until time $t$, resulting in $\tilde{\mathbf{x}}(t, -)$ and a model-alone forecast $\mathbf{A}(t)\tilde{\mathbf{x}}(t, -) + \mathbf{B}(t)\tilde{\mathbf{q}}(t)$. Should a measurement $\mathbf{y}(t + \Delta t)$ exist, a weighted average of $\tilde{\mathbf{x}}(t + \Delta t, -)$ and $\mathbf{y}(t + \Delta t)$ provides the "best" estimate, where the relative weighting is by the inverse of their separate uncertainties. In the present case, this best estimate at one time-step in the future is given by,

$\quad \tilde{\mathbf{x}}(t + \Delta t) = \mathbf{A}(t)\tilde{\mathbf{x}}(t, -) + \mathbf{B}(t)\mathbf{q}(t) + \mathbf{K}(t + \Delta t)\left[\mathbf{y}(t + \Delta t) - \mathbf{E}(t + \Delta t)\tilde{\mathbf{x}}(t + \Delta t, -)\right],$ (4)

$\quad \mathbf{K}(t + \Delta t) = \mathbf{P}(t + \Delta t, -)\mathbf{E}(t + \Delta t)^T\left[\mathbf{E}(t + \Delta t)\mathbf{P}(t + \Delta t, -)\mathbf{E}(t + \Delta t)^T + \mathbf{R}(t + \Delta t)\right]^{-1}.$ (5)

As written, this operation is known as the innovation form of the "Kalman filter" (KF), and $\mathbf{K}(t + \Delta t)$ is the "Kalman gain." Embedded in this form are the matrices $\mathbf{P}(\mathbf{t} + \mathbf{\Delta t}, -)$ and $\mathbf{R}(t + \Delta t)$ which denote the uncertainty of the pure model prediction at time $t$ and the covariance of the observation noise (usually assumed to have zero mean error) respectively. The operators

$\mathbf{P}(\mathbf{t}, -)$ and $\mathbf{P}(\mathbf{t})$ evolve with time according to a matrix Riccati equation; see Appendix (A1), Wunsch (2006) or numerous textbooks (e.g. Stengel (1986), Goodwin and Sin (1984)) for a fuller discussion. Although possibly looking unfamiliar, Eq. (4) is simply a convenient rewriting of the matrix-weighted average of the model forecast at $t + \Delta t$ with that determined from the data. If no data exist, $\mathbf{y}(t + \Delta t) = \mathbf{E}(t + \Delta t) = 0$, and the system reduces to the ordinary model prediction.

Without doing any calculations, some surmises can be made about system behavior from Eq. (4). Among them are: (a)

If the initial condition (with uncertainty $\mathbf{P}(0)$) has errors, the time evolution will propagate initial condition errors forward. Similarly, however obtained, any error in $\tilde{\mathbf{x}}(t, -)$ with uncertainty covariance $\mathbf{P}(t + \Delta t, -)$ will be propagated into $\tilde{\mathbf{x}}(t + \Delta t)$. (b) Importance of the data versus the model evolution depends directly on the ratio of the norms of $\mathbf{E}(\tau)\mathbf{P}(\tau, -)\mathbf{E}(\tau)^T$, $\mathbf{R}(\tau)$. Lastly (c), most important for this paper, the data disturbances appear in the time-evolution equation (4) fully analogous to the external source-sink/boundary condition term. Conservation laws implicit in the model-alone will be violated in the time-

evolution, and ultimately methods to obviate that problem must be found.

Although here the true Kalman filter is used for toy models to predict time series, in large-scale ocean or climate models such is almost never the case in practice. Calculation of the $\mathbf{P}$ matrices from Eq. (A1) is computationally so burdensome that $\mathbf{K}(t)$ is replaced by some very approximate or intuitive version, usually constant in time, potentially leading to further major errors beyond what is being discussed here.

## 3 Example 1: Mass-Spring Oscillator

Consider the intuitively accessible system of a mass-spring oscillator, following any of McCuskey (1959), Goldstein (1980), Wunsch (2006), Strang (2007), initially in the conventional continuous time formulation of simultaneous differential equations. Three identical masses, $m = 1$, are connected to each other and to a wall at either end by springs of identical constant, $k$ (Fig. 1). Movement is damped by a Rayleigh friction coefficient $r$. Generalization to differing masses, spring constants, and dissipation

coefficients is straightforward. Displacements of each mass are $\xi_i(t)$, $i = 1, 2, 3$. The linear Newtonian equations of coupled



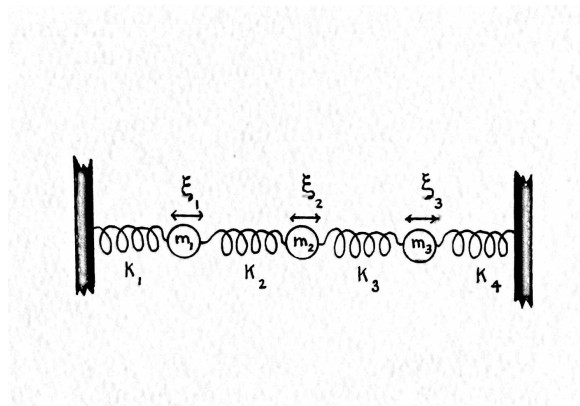

**Figure 1.** Mass-spring oscillator system used as a detailed example. Although the sketch is slightly more general, here all masses have the same value, $m$, and all spring constants and Rayleigh dissipation coefficients $k, r$ are the same.

motion are,

$$m\frac{d^2\xi_1}{dt^2} + k\xi_1 + k(\xi_1 - \xi_2) + r\frac{d\xi_1}{dt} = q_{c1}(t) \tag{6a}$$

$$m\frac{d^2\xi_2}{dt^2} + k\xi_2 + k(\xi_2 - \xi_1) + k(\xi_2 - \xi_3) + r\frac{d\xi_2}{dt} = q_{c2}(t) \tag{6b}$$

$$m\frac{d^2\xi_3}{dt^2} + k\xi_3 + k(\xi_3 - \xi_2) + r\frac{d\xi_3}{dt} = q_{c3}(t). \tag{6c}$$

This second-order system is reduced to a canonical form of coupled first-order equations by introduction of a continuous time state vector, the column vector,

$$\mathbf{x}_c(t) = [\xi_1(t), \xi_2(t), \xi_3(t), d\xi_1/dt, d\xi_2/dt, d\xi_3/dt]^T, \tag{7}$$

where superscript $T$ denotes the transpose. Note the mixture of dimensional units in the elements of $\mathbf{x}_c(t)$, identifiable with the Hamiltonian variables of position and momentum. Then Eqs. (6) become (setting $m = 1$, or dividing through by it),

$$\frac{d\mathbf{x}_c(t)}{dt} = \mathbf{A}_c\mathbf{x}_c(t) + \mathbf{B}_c\mathbf{q}_c(t), \tag{8}$$

where

$$\mathbf{A}_c = \begin{Bmatrix} 0 & 0 & 0 & 1 & 0 & 0 \\ 0 & 0 & 0 & 0 & 1 & 0 \\ 0 & 0 & 0 & 0 & 0 & 1 \\ -2k & k & 0 & -r & 0 & 0 \\ k & -3k & k & 0 & -r & 0 \\ 0 & k & -2k & 0 & 0 & -r \end{Bmatrix} = \begin{Bmatrix} \mathbf{0}_3 & \mathbf{I}_3 \\ \mathbf{K}_c & \mathbf{R}_c \end{Bmatrix}, \tag{9}$$





defining the time-invariant 3x3 block matrices $\mathbf{K}_c$ and $\mathbf{R}_c$, symmetric and diagonal respectively. The structure of $\mathbf{B}_c$ depends on which masses are forced. For example, if only $\xi_2(t)$ is forced, then $\mathbf{B}_c$ would be the unit vector in the second element.

Assuming $r$, $k \neq 0$, $\mathbf{A}$ is full-rank with three pairs of complex conjugate eigenvalues, but non-orthonormal right eigenvectors. $\mathbf{A}$ and $\mathbf{B}$ are both assumed time-independent. Discussion of the physics and mathematics of small oscillations can be found in most classical mechanics textbooks and is omitted here. What follows is left in dimensional form to make the results most intuitively accessible.

*Energy*

Consider now an energy principle. Let $\boldsymbol{\xi} = (\xi_1(t), \xi_2(t), \xi_3(t))^T$ be the position sub-vector. Define, without dissipation ($\mathbf{R}_c = \mathbf{0}$) or forcing,

$$\mathcal{E}_c(t) = \frac{1}{2}\left[\left(\frac{d\boldsymbol{\xi}}{dt}\right)^T\left(\frac{d\boldsymbol{\xi}}{dt}\right) - \boldsymbol{\xi}^T\mathbf{K}_c\boldsymbol{\xi}\right] \tag{10}$$

$$\frac{d\mathcal{E}_c(t)}{dt} = \frac{1}{2}\frac{d}{dt}\left[\left(\frac{d\boldsymbol{\xi}}{dt}\right)^T\left(\frac{d\boldsymbol{\xi}}{dt}\right) - \boldsymbol{\xi}^T\mathbf{K}_c\boldsymbol{\xi}\right] = 0. \tag{11}$$

Here $\mathcal{E}_c$ is the sum of the kinetic and potential energies (the minus sign compensates for the negative definitions in $\mathbf{K}_c$). The

non-diagonal elements of $\mathbf{K}_c$ redistribute the potential energy amongst the masses through time.

With finite dissipation and forcing,

$$\frac{d\mathcal{E}_c(t)}{dt} = \left(\frac{d\boldsymbol{\xi}}{dt}\right)^T\mathbf{R}_c\left(\frac{d\boldsymbol{\xi}}{dt}\right) + \frac{d\boldsymbol{\xi}}{dt}^T\mathbf{B}_c\mathbf{q}(t). \tag{12}$$

If the forcing and dissipation vanish then $d\mathcal{E}_c(t)/dt = 0$ (see Cohn (2010) for a formal discussion of such continuous time systems.)

*Discretization*

Eq. (1) is discretized at time intervals $\Delta t$ using an Eulerian time-step in the same form,

$$\mathbf{x}(t+\Delta t) = \mathbf{A}\mathbf{x}(t) + \mathbf{B}\mathbf{q}(t), \quad t = n\Delta t, \tag{13}$$

for $n \geq 0$. The prediction model is unchanged except now,

$$\mathbf{A} = \begin{Bmatrix} 1 & 0 & 0 & \Delta t & 0 & 0 \\ 0 & 1 & 0 & 0 & \Delta t & 0 \\ 0 & 0 & 1 & 0 & 0 & \Delta t \\ -2k\Delta t & k\Delta t & 0 & 1-r\Delta t & 0 & 0 \\ k\Delta t & -3k\Delta t & k\Delta t & 0 & 1-r\Delta t & 0 \\ 0 & k\Delta t & -2k\Delta t & 0 & 0 & 1-r\Delta t \end{Bmatrix} \tag{14}$$

$$= \begin{Bmatrix} \mathbf{I}_3 & \Delta t\mathbf{I}_3 \\ \Delta t\mathbf{K}_c & \mathbf{I}_3 + \Delta t\mathbf{R}_c \end{Bmatrix} \tag{15}$$





and without the $c$ subscript.

For this choice of the discrete state vector, the energy rate of change is formally analogous to that in the continuous case,

$$\frac{\mathcal{E}(t) - \mathcal{E}(t - \Delta t)}{\Delta t} = \left(\frac{d\boldsymbol{\xi}}{dt}\right)^T \mathbf{R}\left(\frac{d\boldsymbol{\xi}}{dt}\right) + \frac{d\boldsymbol{\xi}}{dt}^T \mathbf{B}\mathbf{q}(t) \tag{16}$$

where $\mathcal{E}(t)$ is computed as before in Eq. (10) except now using the discretized $\mathbf{x}(t)$. An example solution for the nearly
dissipationless, unforced oscillator is provided in Fig. 2, produced by the discrete formulation $\mathcal{E}(t)$. The potential and kinetic energies through time are shown in Fig. 2c, along with elements and derived quantities of the state vector in Fig. 2a, 2b. Non-zero values here arise only from the initial condition $\mathbf{x}(0) = [1, 0, 0, 0, 0, 0]^T$, and necessarily involve specifying both positions and their rates of change. A small amount of dissipation ($r = 0.5$) was included to stabilize the particularly simple numerical scheme. The basic oscillatory nature of the state vector elements is plain, and the decay time is also visible.

The total energy declines over the entire integration time, but with small oscillations persisting after 5000 time steps. Kinetic energy is oscillatory as energy is exchanged with the potential component.

## 3.1 Mass-Spring Oscillator with Observations

Note that if the innovation form of the evolution, Eq. (4), is used, the energy change becomes,

$$\frac{\mathcal{E}(t) - \mathcal{E}(t - \Delta t)}{\Delta t} \approx \left(\frac{d\boldsymbol{\xi}}{dt}\right)^T \mathbf{R}\left(\frac{d\boldsymbol{\xi}}{dt}\right) + \frac{d\boldsymbol{\xi}}{dt}^T \mathbf{B}\mathbf{q}(t) + \frac{d\mathbf{x}(t)}{dt}^T \mathbf{K}(t)\left[\mathbf{y}(t) - \mathbf{E}(t)\mathbf{x}(t)\right],$$

showing explicitly the influence of the observations. With intermittent observations and/or with changing structures in $\mathbf{E}(t)$, then $\mathcal{E}(t)$ will undergo forced abrupt changes that are a consequence of the sequential innovation.

Given the very large number of potentially erroneous elements in any choice of model, data and data distributions, and the ways in which they interact when integrated through time, a comprehensive discussion even of the 6-element state vector mass-spring oscillator system is difficult. Instead, some simple examples exploring primarily the influence of data density on
the state estimate and of its mechanical energy are described. Numerical experiments are readily done with the model and its time-constants, model time-step, accuracies and corresponding covariances of initial conditions, boundary conditions, data etc. The basic problems of any linear or linearized system already emerge in this simple example.

The "true" model assumes the parameters $k = 30$, $r = 0.5$, and $\Delta t = .001$, and is forced by

$$\mathbf{q}(t) = q_1(t) = 0.1\cos[2\pi t/(2.5\, T_{diss})] + \varepsilon(t). \tag{17}$$

That is, only mass one is forced in position, and with a low frequency not equal to one of the natural frequencies. In this case, $\mathbf{B} = [1, 0, 0, 0, 0, 0]^T$ and $\mathbf{q}(t) = q_1(t)$, a scalar. The dissipation time is $T_{diss} = 1/r$, and $\varepsilon(t)$ is a white noise element with standard deviation 0.1. Initial condition is $\xi_1(0) = 1$, all other elements vanishing; see Fig. 3 for an example of a forced solution of positions, velocities, and derived quantities. Accumulation of the influence of the stochastic element in the forcing depends directly upon details of the model time-scales, and, if $\varepsilon(t)$ were not white noise, on its spectrum as well. In all cases
the cumulative effect of a random forcing will be a random walk—with details dependent upon the forcing structure, as well as on the various model time scales.

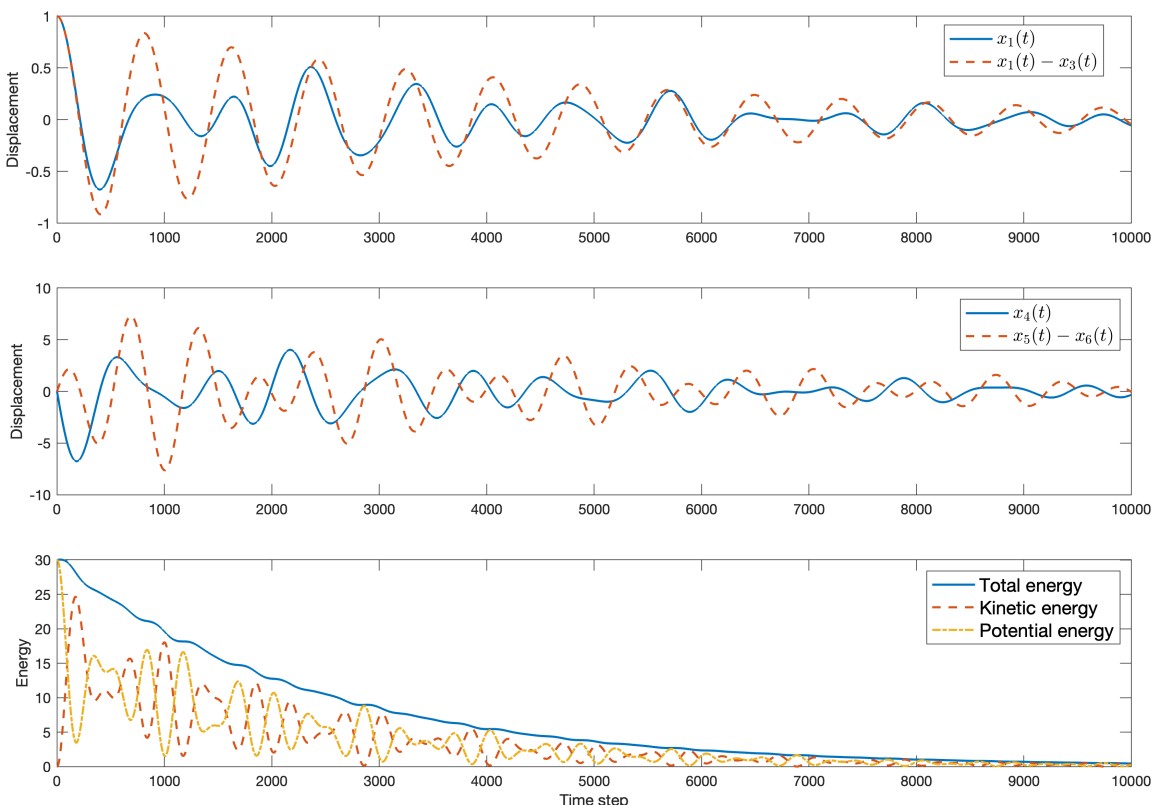

**Figure 2.** The unforced case, initial condition vanishing except for $\xi_1(1) = 1$. Natural frequency and decay scale are apparent. (a) $x_1(t)$ (solid) and $x_1(t) - x_3(3)$ (dashed). (b) $x_4$ (solid) and $x_5 - x_6$ (dashed). (c) $\mathcal{E}(t)$ showing decay scale from the initial displacement, alongside kinetic energy (dashed) and potential energy (dot-dashed.)

The prediction model here has fully known initial conditions and $\mathbf{A}, \mathbf{B}$ matrices, but the stochastic component of the forcing is being treated as fully unknown, i.e., $\varepsilon(t) = 0$. Added noise in the data values has a standard deviation of $0.01$ in all calculations. The numerical experiments and their parameters are defined in Table 1.

The experiments and their parameters are outlined in Table 1, where " $-$ " is used to indicate that the same conditions as the nominal truth are used:



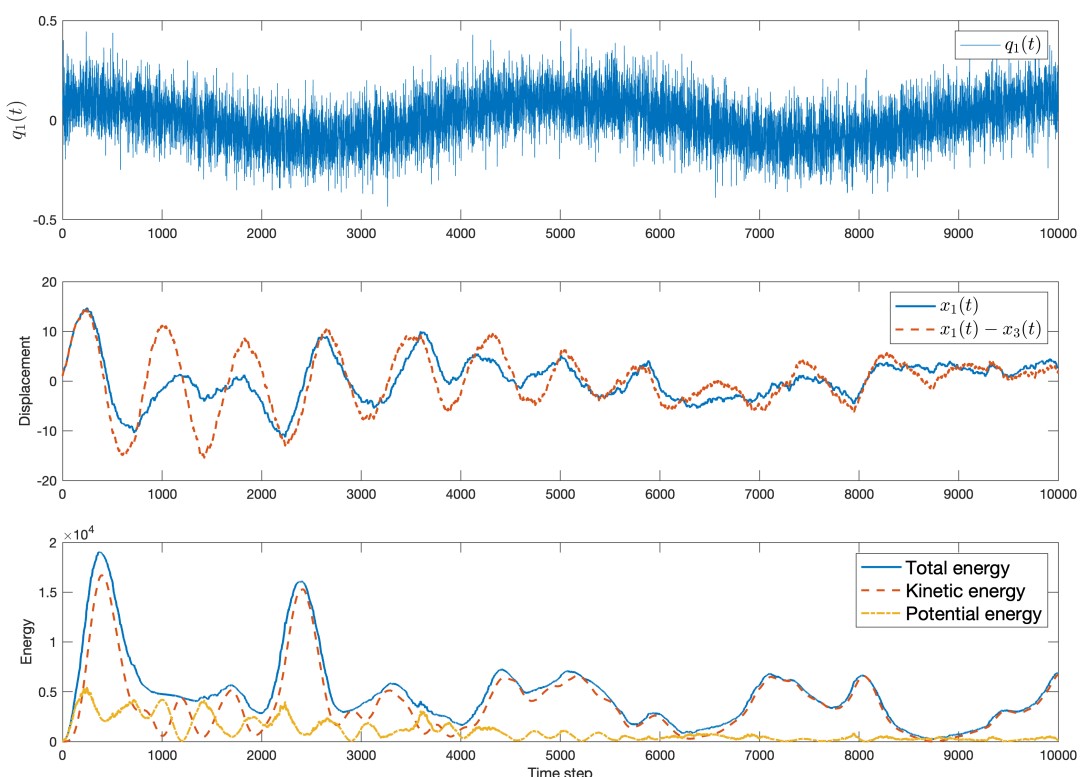

**Figure 3.** Forced version of the same oscillator system as in Fig. 2. Forcing is at every time-step in the mass one position alone. (a) The forcing, $\mathbf{q}(t)$, is given by white noise plus the visible low frequency cosine curve. (b) $x_1(t)$ (solid) and $x_1(t) - x_3(t)$ (dashed). (c) Total energy through time, $\mathcal{E}(t)$ alongside kinetic energy (dashed) and potential energy (dot-dashed). Energy varies with the random walk arising from $\varepsilon(t)$ as well as from the deterministic forcing.

| | 3.1.1: Accurate observations at two timesteps and multiple timesteps | 3.1.2: Fixed position | 3.1.3: Observations of averages |
|---|---|---|---|
| Nominal truth | Forcing given by $\mathbf{q}(t)$ (Eq. (17)), observational noise is $\sigma(\varepsilon(t)) = 0.1$, | $\mathbf{x}_0 = (1, 0, 2, 0, 0, 0)$ All else the same as 3.1.1 | Same as 3.1.1 |
| Prediction | Forcing given by $0.5\,\mathbf{q}(t) - \varepsilon(t)$ | – | – |
| Kalman filter | Forcing given by $0.5\,\mathbf{q}(t) - \varepsilon(t)$, observational noise is $\sigma = 0.01$ | – | – |

**Table 1.** Numerical experiments and corresponding parameters. Common to all configurations are the following settings: $k = 30$, $r = 0.5$, $\Delta t = 0.001$, $\mathbf{x}_0 = \mathbf{e}_1$.





### 3.1.1 Accurate Observations: Two Times and Multiple Times

To demonstrate the most basic problem of estimating energy, consider highly accurate observations of all six generalized
coordinates (i.e., positions and velocity) at two times $\tau_1, \tau_2$ as displayed in Fig. 4 with $\mathbf{E} = \mathbf{I}_6$, i.e. having no observational null-
space. The forecast model has the correct initial conditions of the true state but incorrect forcing: the deterministic component
has half the amplitude of the true forcing and $\varepsilon(t)$ is completely unknown. Noise with standard deviation $0.01$ is added to
the observations. Although the new estimate of the KF reconstructed state vector is an improvement over that from the pure
forecast, any effort to calculate a true trend in the energy of the system, $\tilde{\mathcal{E}}(t)$, will fail unless careful attention is paid to
correcting for the conservation violations at the times of the observation.

Until the first data point (a vertical line in Fig. 4) is introduced, the KF and prediction energies are identical, as expected.
Energy discontinuities occur at each introduction of a data point ($t = 5000\Delta t$ and $t = 7300\Delta t$). After the first data point the KF
energy remains lower than the true energy, but the KF prediction is nonetheless an improvement over that from the prediction
model alone. Even if the observations are made perfect ones (not shown), this bias error in the energy persists (see e.g., Dee
(2005)). Figure 4c offers insight into the KF prediction via the covariance matrix $\mathbf{P}(t)$.

Figure 5 shows the results when observations occur in clusters having different intervals between the measurements; the
first being sparser observations (300 timesteps between data points) and the second being denser observations (125 timesteps
between data points). Visually, the displacement and energy have a periodicity imposed by the observation time-intervals and
readily confirmed by Fourier analysis. Again, the KF solution is the pure model prediction until data are available, at which
point multiple discontinuities occur, one for every $t$ where data are introduced.

A great many further specific calculations can provide insight as is apparent in the above examples, and as inferred from
the innovation equation. For example, the periodic appearance of observations introduces periodicities in $\tilde{\mathbf{x}}(t)$, and hence in
properties such as the energy derived from it. Persistence of the information in these observations at future times will depend
upon model time-constants including dissipation rates.

### 3.1.2 A Fixed Position

Exploration of the dependencies of energies of the mass-spring system is interesting and a great deal more can be said. Turn
however, to a somewhat different invariant: suppose that one of the mass positions is fixed, but with the true displacement
unknown to the analyst. A significant literature exists devoted to finding changes in scalar quantities such as global mean
atmospheric temperatures, or oceanic currents, with the Atlantic Meridional Overturning Circulation (AMOC) being a favorite
focus. These quantities are typically sub-elements of complicated models involving very large state vectors.

The true model is now adjusted to include the constraint that $x_3(t) = \xi_3(t) = 2$ and thus $x_6(t) = d\xi_3(t)/dt = 0$. That is,
a fixed displacement in mass 3 (and consequently a zero velocity in mass 3) is used in computing the true state vector. The
forecast model has the correct initial condition and incorrect forcing: with a deterministic component again having half the
amplitude and fully unknown noise $\varepsilon(t)$. Observations are assumed to be those of all displacements and velocities, with the
added noise having standard deviation $0.01$. Results are shown in Fig. 6. The question is whether one can infer accurately



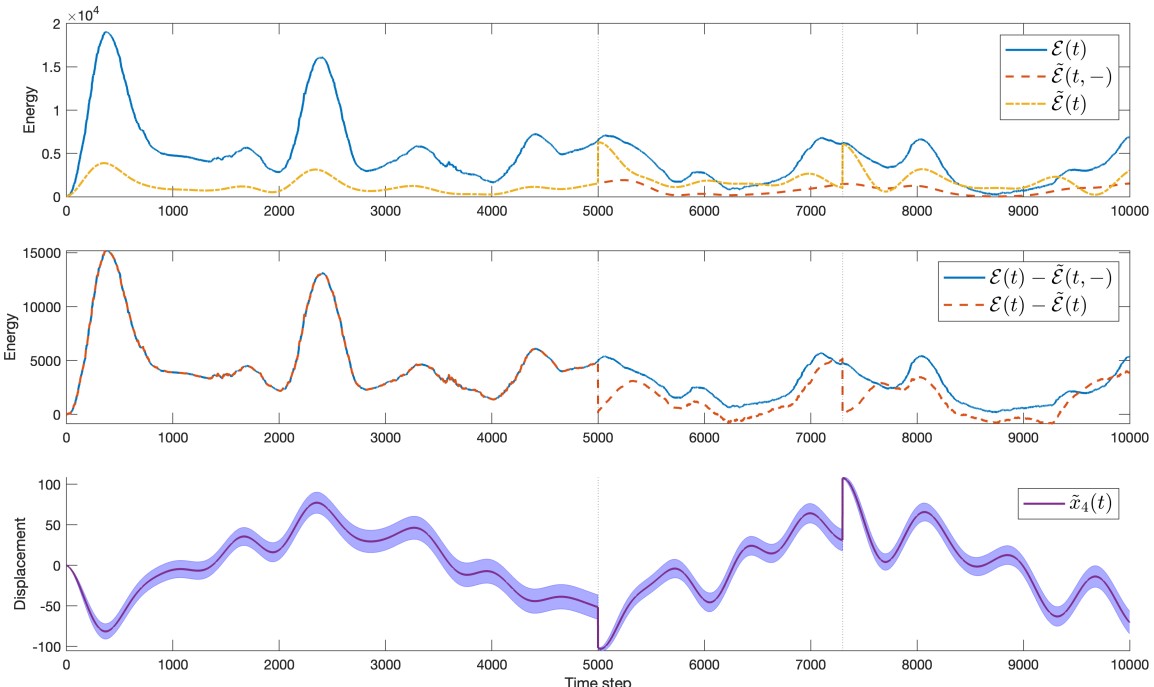

**Figure 4.** (a) Total energy for the 3-mass-spring oscillator system (solid), $\mathcal{E}(t)$, the prediction model (dashed), $\tilde{\mathcal{E}}(t,-)$, and for the KF reconstruction (dot-dashed), $\tilde{\mathcal{E}}(t)$. (b) The difference between truth and prediction total energies (solid) and the difference between truth and KF total energies (dashed). The data points create discontinuities, forcing $\mathcal{E}(t) - \tilde{\mathcal{E}}(t)$ to be near zero. Note that the differences do not perfectly match at time $t = 0$, but they are relatively small. (c) Estimated position for velocity in the first mass ($d\xi_1/dt = x_4(t)$) from the Kalman filter, showing the jump at the two times where there are complete near-perfect data. Standard error bar is shown from the corresponding diagonal element of $\mathbf{P}(t)$, in this case given by $\sqrt{P_{44}}$.

that $\xi_3(t)$ is a constant through time. A KF estimate for the fixed position, $\tilde{x}_3(t) = \tilde{\xi}_3(t)$, is shown in Fig. 6(a) and includes a substantial error in its value (and its variations or trends) at all times. Exceptions occur when data are introduced at the vertical lines in Fig. 6(a), (b). Owing to the noise in the observations, the KF cannot reproduce a perfect result.

Estimated mass 3 position variations occur even during the data dense period and arise both from the entry of the data and the noise in the observations. An average taken over the two-halves of the observation interval might lead to the erroneous conclusion that a decrease had taken place. Such an incorrect inference can be precluded by appropriate use of computed uncertainties. Note also the impact of the KF on the energy (Fig. 6c,d), producing artificial changes as in the previous experiment.



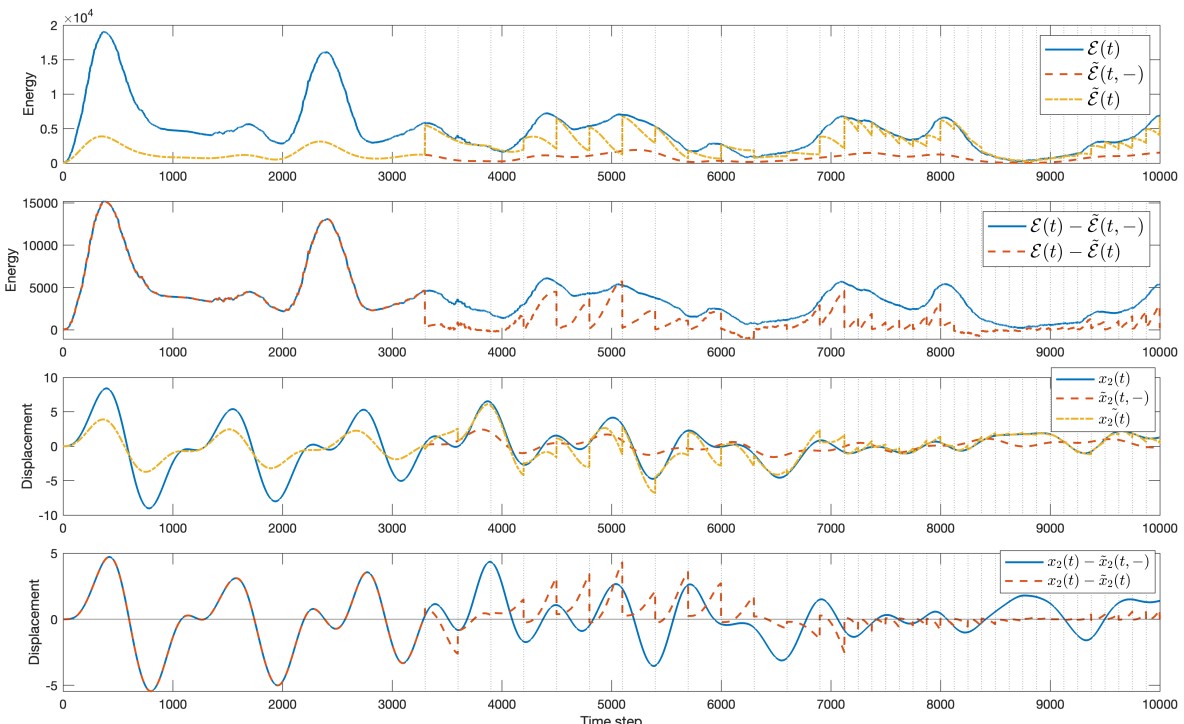

**Figure 5.** (a) Similar to Fig. 4. Shown are true total energy (solid), $\mathcal{E}(t)$, plotted alongside the prediction model energy (dashed), $\tilde{\mathcal{E}}(t, -)$, and calculated from KF algorithm (dot-dashed), $\tilde{\mathcal{E}}(t)$. (b) The difference between the "truth" and prediction (solid) alongside the difference between "truth" and KF (dashed). (c) The position of mass two, $x_2(t)$, given by the "true" model (solid), the prediction (dashed), and the KF estimate (dot-dashed). The introduction of data points forces the KF to match the state vector from the data, creating the discontinuities expected. (d) $x_2(t) - \tilde{x}_2(t, -)$ (solid) alongside $x_2(t) - \tilde{x}_2(t)$ (dashed).

### 3.1.3 Observations of Averages

Consider now a set of observations of the average of the position of masses 2 and 3, and of the average velocity of masses 1 and 2, mimicking the type of observations that might be available in a realistic setting. Again for optimistic simplicity, the observations are relatively accurate (including noise with standard deviation 0.01) and occur in the two different sets of periodic time intervals. Prediction begins with the correct initial conditions, and again the forcing has half the correct amplitude with fully unknown random forcing. Figure 7 displays the results. Position estimates shown are good, but not perfect, as is also true for the total energy. The energy estimate carries oscillatory power with the periodicity of the oncoming observation intervals and appears in the spectral estimate (not shown) with excess energy in the oscillatory band, and somewhat too low energy at the longest periods. Irregular observation spacing would generate a more complicated spectrum in the result.



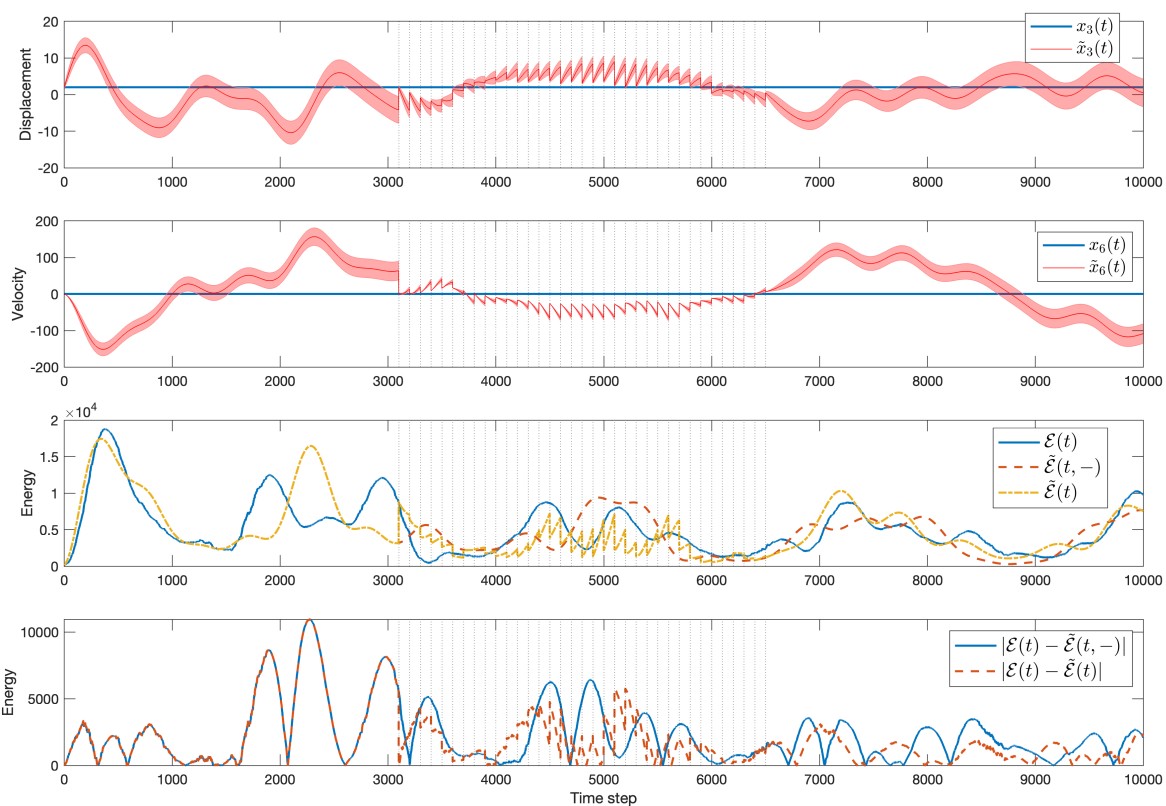

**Figure 6.** (a) Correct value of the constant displacement $x_3(t)$ (solid line) and the estimated value from the KF calculation (dot-dashed line) with error bar computed from $\mathbf{P}$. Vertical lines are again the observation times. (b) Correct value of the constant velocity $x_6(t)$ (solid line) and the estimated value from the KF calculation (dot-dashed line) with error bar computed from $\mathbf{P}$. (c) The total energy given by the true model (solid line), the prediction value (dashed line), and the KF estimate (dot-dashed line.) (d) The absolute value of the difference between truth and prediction (solid line), and the absolute value of the difference between truth and the KF value (dashed line.)



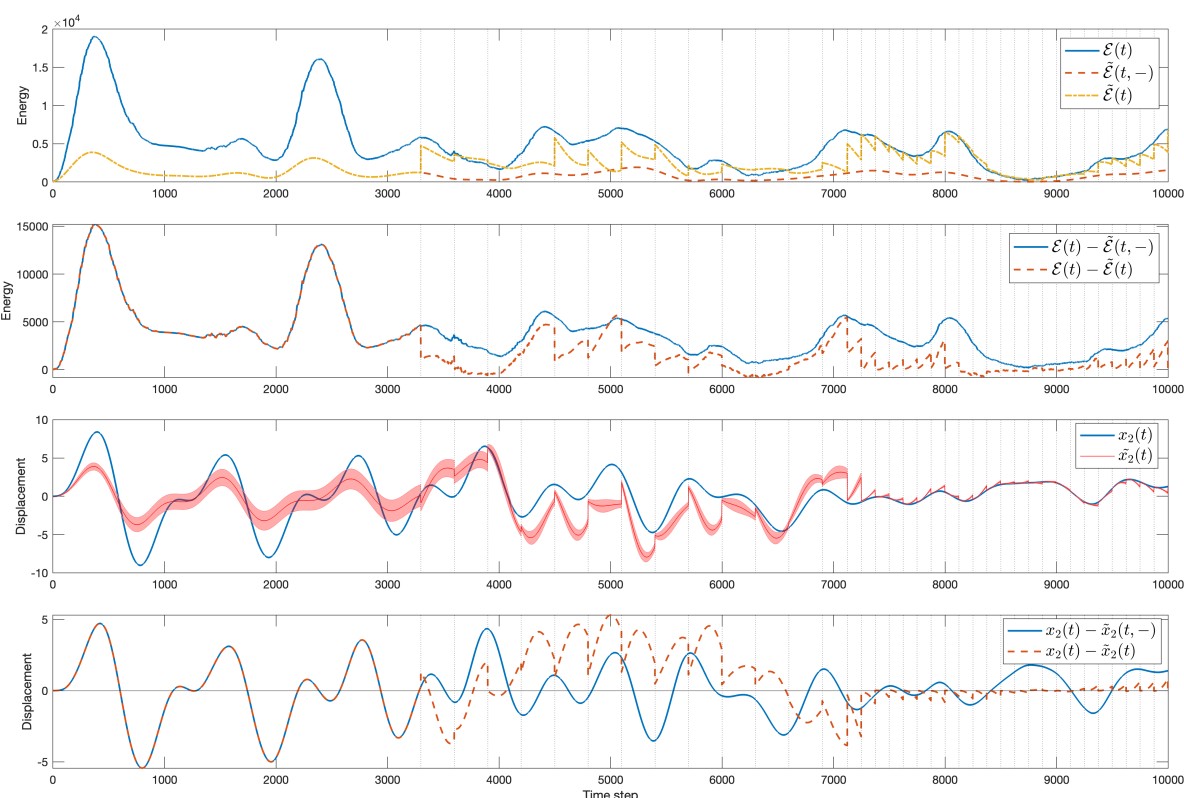

**Figure 7.** (a) Results for total energy when observations were of the average of the two positions $x_2(t), x_3(t)$ and the two velocities, $x_4(t), x_5(t)$ at the times marked by vertical lines. (b) Total energy differences corresponding to the situation in (a). (c) Results for the displacement $x_2(t)$ estimate when observations were of averages along with error bar from $\mathbf{P}$. (d) Differences between true and predicted (solid blue) and true and KF (dashed red).





A general discussion of nullspaces involves that of the column weighted $\mathbf{P}\left(\tau,-\right)\mathbf{E}^T$ appearing in the Kalman gain. If $\mathbf{E}$ is the identity (i.e. observations of all positions and velocities), and $\mathbf{R}\left(\tau\right)$ has sufficiently small norm, all elements of $\mathbf{x}\left(\tau\right)$ are resolved. In the present case, with $\mathbf{E}$ having two rows, corresponding to observations of the averages of two mass positions and

of two velocity positions, the resolution analysis is more structured than the identity with,

$$\mathbf{E}=\left\{\begin{array}{cccccc} 0 & 1/2 & 1/2 & 0 & 0 & 0 \\ 0 & 0 & 0 & 1/2 & 1/2 & 0 \end{array}\right\}. \tag{18}$$

A singular value decomposition $\mathbf{E}=\mathbf{U}\mathbf{S}\mathbf{V}^T=\mathbf{U}_2\mathbf{S}_2\mathbf{V}_2^T$, produces two non-zero singular values, where $\mathbf{U}_2$, etc. denotes the first two columns of the matrix. At rank 2, the resolution matrices, $\mathbf{T}_U$, $\mathbf{T}_V$, based on the $\mathbf{U}$, $\mathbf{V}$ vectors respectively and the standard solution covariances are easily computed (Wunsch (2006)). $\mathbf{A}$ distributes information about the partially determined

$x_i$ throughout all masses via the dynamical connections as contained in $\mathbf{P}\left(\tau\right)$. Bias errors require specific, separate analysis.

The impact of an observation on future estimated values tends to decay in time, dependent upon the model time-scales. Insight into the future influence of an observation can be obtained from the Green function discussion in the Appendix.

## 3.2 Uncertainties

In a linear system, a Gaussian assumption for the dependent variables is commonly appropriate. Here the quadratic dependent

energy variables become $\chi^2$ distributed. Thus the $\xi_i^2,\dot{\xi}_i^2$ have such distributions, but with differing means and variances, and with potentially very strong correlations, so that they cannot be regarded as independent variables. Determining the uncertainties of the six covarying elements making up $\tilde{\mathcal{E}}\left(t\right)$ involves some intricacy. A formal analysis can be made of the resulting probability distribution for the sum in $\tilde{\mathcal{E}}\left(t\right)$, involving non-central $\chi^2$ distributions (Imhof (1961), Sheil and O'Muircheartaigh (1977), Davies (1980)). As an example, an estimate of the uncertainty could be made via a Monte Carlo approach by generating

$N$ different versions of the observations, differing in the particular choice of noise value in each and tabulating the resulting range. These uncertainties can be used to calculate, e.g., the formal significance of any apparent trend in $\tilde{\mathcal{E}}\left(t\right)$. Implicit in such calculations is having adequate knowledge of the probability distribution from which the random variables are obtained. An important caveat is that bias errors such as those seen in the energy estimates in Fig. 7 must be separately determined.

The structure of the uncertainty operator $\mathbf{P}$ depends upon both the model and the detailed nature of the observations via

Eq. (A3). Suppose observations only provide knowledge of the velocity of mass 2, $x_5(t)$. Consider $\mathbf{P}(t=7124)$, just before observations become available (i.e., the model has mimicked a true prediction until this point), and $\mathbf{P}(t=8250)$, after ten observations of $x_5$ have been incorporated with the Kalman filter. The resulting $\mathbf{P}\left(t\right)$ following the observations produces highly inhomogeneous variances (the diagonals of $\mathbf{P}$). In this particular case, one of the eigenvalues of $\mathbf{P}\left(\tau\right)$ for $\tau$ just beyond the time of any observation, is almost zero, meaning that $\mathbf{P}(\tau)$ is nearly singular (Fig. 8). The corresponding eigenvector

has a value near 1 in position 5 and is near zero elsewhere. Because numerous accurate observations were made of $x_5(t)$, its uncertainty almost vanishes for that element, and a weighting of values by $\mathbf{P}\left(t\right)^{-1}$, gives it a near infinite weight at that time.





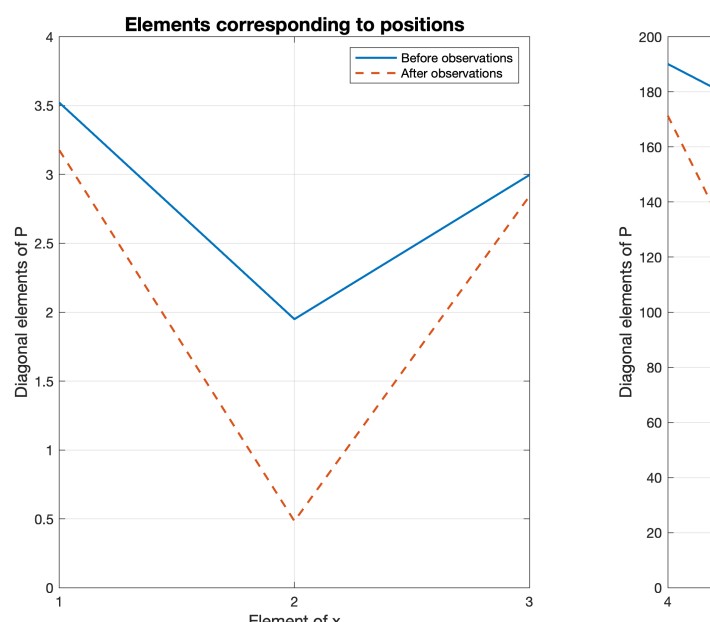
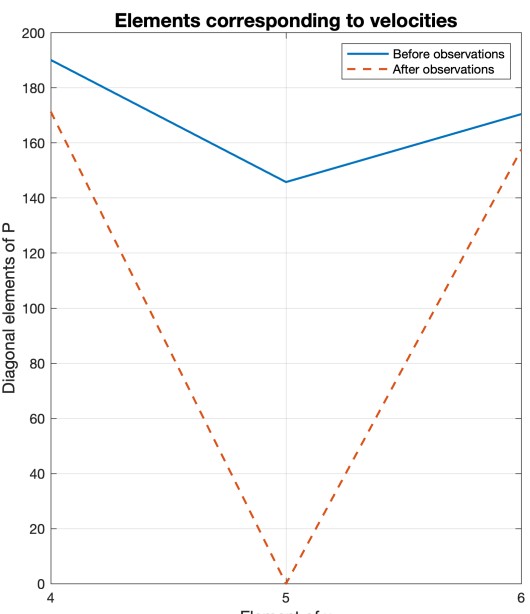

**Figure 8.** From left to right: (a) The first three diagonal elements of $\mathbf{P}$ before any observations of $x_5$ (solid) and after ten observations of $x_5$ (dashed). (b) The last three diagonal elements of $\mathbf{P}$ before any observations (solid) and after ten observations (dashed.)

### 3.3 A Fixed-Interval Smoother

The Kalman filter and various approximations to it produce an estimate at any time, $\tau$, taking account only of data at $\tau$ or in the past—with an influence falling as the data recedes into the past, at a rate dependent on the model time scales. But in

many problems, such as those addressed (for one example) by the Estimating the Circulation and Climate of the Ocean (ECCO) project, the goal is to find a best-estimate over a finite interval, nominally, $0 \le t \le t_f$, and accounting for all of the observations, whether past or future to any $\tau$. Furthermore, as already noted above, physical sense requires satisfaction of the generalized (to account for sources/sinks) energy, mass, and other important conservation rules. How to do that?

In distinction to the "filtering" goal underlying a KF best-prediction, the fixed interval problem is generally known as that of

"smoothing." Several approaches exist. One of the most interesting, and one leading to the ability to parse data versus model structure impact over the whole interval, is called the Rauch-Tung-Striebel (RTS) smoothing algorithm. In that algorithm, it is assumed that a true KF has already been run over the full time-interval, and that the resulting $\tilde{\mathbf{x}}(t), \mathbf{P}(t), \tilde{\mathbf{x}}(t,-), \mathbf{P}(t,-)$ remain available. The basic idea is subsumed in the algorithm,

$$\tilde{\mathbf{x}}(t,+) = \tilde{\mathbf{x}}(t) + \mathbf{L}(t+\Delta t)\left[\tilde{\mathbf{x}}(t+\Delta t,+) - \tilde{\mathbf{x}}(t+\Delta t,-)\right] \tag{19}$$





with

$$\mathbf{L}(t + \Delta t) = \mathbf{P}(t)\mathbf{A}(t)^T \mathbf{P}(t + \Delta t, -)^{-1} \tag{20}$$

This new estimate, using data *future* to $t$, depends upon a weighted average of the previous best estimate $\tilde{\mathbf{x}}(t)$ with its difference

between the original pure prediction, $\tilde{\mathbf{x}}(t + \Delta t, -)$ and the improvement (if any) made of the later estimate at $t + \Delta t$. The

latter uses any data that occured *after* that time. Thus a backwards-in-time recursion of Eq. (19) is done—starting from the

best estimate at the final KF time, $t = t_f$, beyond which no future data occur. The RTS coefficient matrix, $\mathbf{L}(t + \Delta t)$, has a

particular structure accounting for the correlation between $\tilde{\mathbf{x}}(t + \Delta t, +)$ and $\tilde{\mathbf{x}}(t + \Delta t, -)$ generated by the KF. Equation (A5)

calculates the new uncertainty, $\mathbf{P}(t, +)$.

In this algorithm a correction is also necessarily made to the initial assumptions concerning $\mathbf{q}(t)$, producing a new set of

vector forcings, $\tilde{\mathbf{q}}(t, +) = \mathbf{q}(t) + \tilde{\mathbf{u}}(t)$ such that the new estimate, $\tilde{\mathbf{x}}(t, +)$ exactly satisfies Eq. 1 with $\tilde{\mathbf{q}}$, at *all* times, over the

interval. If the true model satisfies energy, so will the new estimate. Estimated $\tilde{\mathbf{u}}(t, +)$, often called the "control vector," has its

own computable uncertainty found from Eq. (A5b). In many problems, an improved knowledge of the forcing field/boundary

conditions may be equally or more important than is improvement of the state vector. Application of the RTS algorithm is

made in the following section to a slightly more geophysical example.

## 4 Example 2: Barotropic Rossby Waves

Consider the smoothing problem in a geophysical fluid dynamics toy model. Realism is still not the goal, which remains as the

demonstration of various elements making up estimates in simplified settings.

### 4.1 Rossby Wave Normal Modes

A flat-bottom, linearized $\beta$-plane Rossby wave system, has a two-dimensional governing equation for the streamfunction, $\psi$,

$$\frac{\partial \nabla^2 \psi}{\partial t} + \beta \frac{\partial \psi}{\partial x} = 0, \tag{21}$$

in a square beta-plane basin of horizontal dimension $L$. This problem is representative of those involving both space and time

structures, including boundary conditions. (Spatial variables $x, y$ should not be confused with the state vector or data variables).

Eq. (21) and other geophysically important ones are not self-adjoint, and the general discussion of quadratic invariants leads

inevitably to adjoint operators (see Morse and Feshbach (1953) or for bounding problems—Sewell et al. (1987), Chs. 3, 4).

The closed-basin problem was considered by Longuet-Higgins (1964). Pedlosky (1965) and LaCasce (2002) provide helpful

discussions of normal modes. Relevant real observational data are discussed by Luther (1982), Woodworth et al. (1995), Ponte

(1997), Thomson and Fine (2021) and others. The domain here is $0 \leq x \leq L$, $0 \leq y \leq L$ with boundary condition $\psi = 0$ on all

four boundaries.





Introduce non-dimensional primed variables $t' = ft$, $x = Lx'$, $y = Ly'$, $q = q_0 q'$, and $\psi' = (a^2/f)\psi$. Letting $a$ be the radius of the Earth, and $\beta = \beta' f/a = 1.7$, Eq. (21) is non-dimensionalized as

$$\frac{\partial \nabla'^2 \psi'}{\partial t'} = \beta' \frac{L}{a} \frac{\partial \psi'}{\partial x'} = 0. \tag{22}$$

Choosing further $L = a$, and then omitting the primes from here on except for $\beta'$,

$$\frac{\partial \nabla^2 \psi}{\partial t} + \beta' \frac{\partial \psi}{\partial x} = 0 \tag{23}$$

Haier et al. (2006) describe numerical solution methods that specifically conserve invariants, but these methods are not used here. Gaspar and Wunsch (1989) employed this system for a demonstration of sequential estimation with altimetric data. Here a different state vector is used.

An analytical solution to (23) is,

$$\psi(t, x, y) = \sum_{n=0}^{N} \sum_{m=0}^{M} \exp(-i\sigma_{nm} t) c_{nm} e^{-i\beta' x/\sigma_{nm}} \sin(n\pi x) \sin(m\pi y), \tag{24}$$

along with the dispersion relation,

$$\sigma_{nm} = -\frac{\beta'/2}{\sqrt{(n\pi/L)^2 + (m\pi/L)^2}} \tag{25}$$

where $c_{nm}$ is a coefficient dependent only upon initial conditions in the unforced case.

A state vector is then

$$\mathbf{x}(t) = \{c_p(t)\},$$

where $p$ is a linear ordering of $n, m$. Total dimension is then $N \cdot M$, with $N, M$ the upper limits in Eq. (24). State transition can be written in the now familiar form,

$$x_j(t + \Delta t) = \exp(-i\sigma_j \Delta t) x_j(t), \quad j = 1, .., NM. \tag{26}$$

For numerical examples with the KF and RTS smoother, a random forcing $q_j(t)$ is introduced at every step so that,

$$x_j(t + \Delta t) = \exp(-i\sigma_j \Delta t) x_j(t) + q_j(t), \quad j = 1, .., NM. \tag{27}$$

Note that with the introduction of a forcing these coefficients do not strictly satisfy Eq. (23) but are rather an over-simplified version of a forced solution. Discussion of this dynamical system is still useful for understanding the difficulties that arise in numerical data assimilation. An example of a true forced solution to (21) is examined in Pedlosky (1965).

The problem is now made a bit more interesting by addition to $\psi_1$ of a *steady* component, the solution, $\psi_s(x, y)$ from Stommel (1948) whose governing equation is,

$$R_a \nabla^2 \psi_s + \beta \frac{\partial \psi_s}{\partial x} = \sin \pi y, \tag{28}$$





where $R_a$ is a Rayleigh friction.

An approximate solution, written in the simple boundary-layer/interior form is (e.g., Pedlosky (1965)),

$$\psi_s = e^{-x\beta'/R_a'} \sin \pi y + (x-1)\sin \pi y, \tag{29}$$

which leads to a small error in the eastern boundary condition (numerical calculations that follow used the full Stommel (1948) solution). The $\sin \pi y$ arises from Stommel's assumed time-independent wind-curl.

The new state vector becomes,

$$\mathbf{x}(t) = \begin{Bmatrix} c_p(t) \\ 1 \end{Bmatrix} \tag{30}$$

now of total dimension $N \cdot M + 1$.

The state transition matrix $\mathbf{A}$ is diagonal with the first $N \cdot M$ diagonal elements $\mathrm{diag}\left(\exp\left(-i\sigma_j\Delta t\right)\right)$ and $A_{N\cdot M+1,N\cdot M+1} = 1$, square of dimension $N \cdot M + 1$. A small, numerical dissipation is introduced, multiplying $\mathbf{A}$ by $\exp(-b)$ for $b > 0$, to accommodate loss of memory, e.g., as a conventional Rayleigh dissipation. The operator $\mathbf{B}$ is diagonal with the first $N \cdot$

$M$ diagonal elements $\mathrm{diag}(1)$, and $B_{N\cdot M+1,N\cdot M+1} = 0$ (no forcing is added to the steady solution.) Some special care in computing covariances must be taken when using complex state vectors and transition matrices (Schreier and Scharf (2010)).

Consistent with the analysis in Pedlosky (1965), no westward intensification exists in the normal modes, which decay as a whole. Rayleigh friction of the time-dependent modes is permitted to be different from that in the time-independent mean flow—a physically acceptable situation.

If $\mathbf{q}(t) = 0$ and with no dissipation, then Eq. (23) has several useful conservation invariants including the quadratic invariants of the kinetic energy and of the variance in $\psi$,

$$\Phi(t) = \mathbf{x}(t)^T \mathbf{x}(t) = \sum_{k=1}^{N\cdot M} |x_k(t)|^2 \tag{31}$$

(conjugate transpose); and the linear invariant of the vorticity or circulation—when integrated over the entire basin domain. As above, estimates of the quadratic and linear conservation rules will depend explicitly on initial conditions, forces, distribution

and accuracy of the data, and the covariances and bias errors assigned to all of them.

### 4.1.1 System with Observations

Using the KF plus RTS smoother for sequential estimation, estimates of $\Phi(t)$ as well as the transport of the western boundary current (WBC), $T_{WBC}(t)$, are calculated; the latter is constant in time, although that is unknown to the analyst. Random noise in $T_{WBC}(t)$ exists from both physical noise—the normal modes—and that of the observations $\mathbf{y}(t_j)$, as would be the situation

in nature.

Eq. (1) with the above $\mathbf{A}, \mathbf{B}$ is used to generate the correct fields. Initial conditions, $\mathbf{x}(0)$, in the modal components are,

$$x_p(0) = 1/(n^2 + m^2), \quad p = (n, m), \, n = 3, 4, 5, \, m = 4, 5, \dots, 9 \tag{32}$$



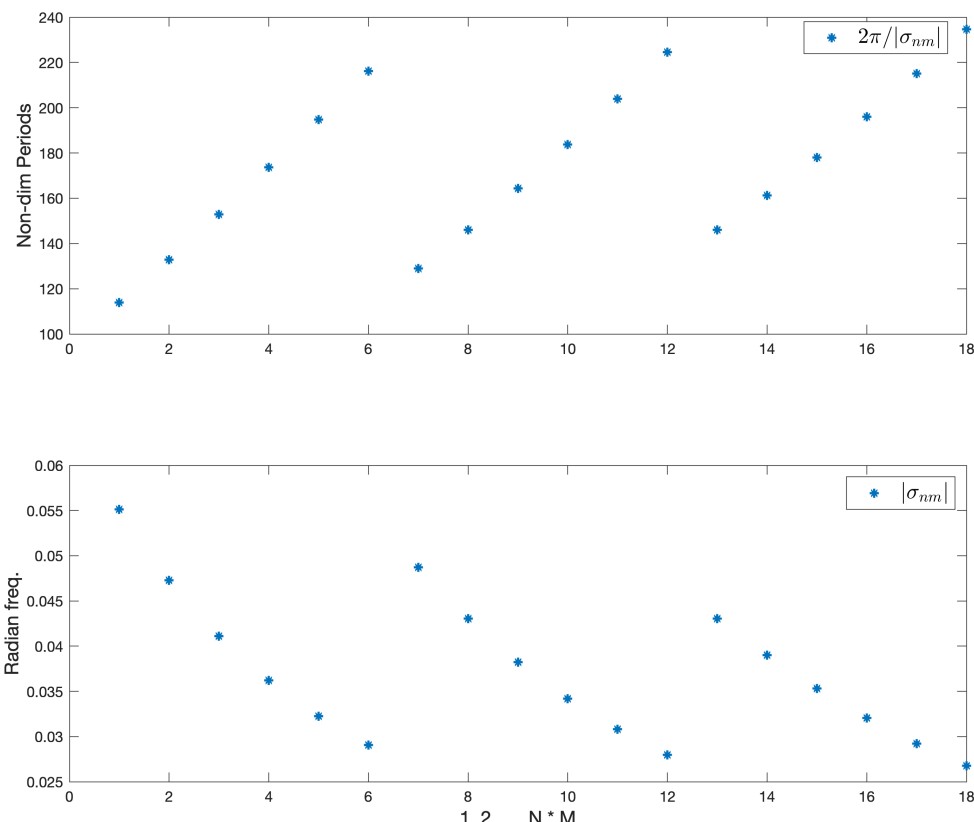

**Figure 9.** (a) Non-dimensional periods. Grouped by fixed $n$ and increasing $m$. (b) Inverse of the periods in (a), giving us the frequencies.

Modal periods are shown in Fig. 9. Parameters are fixed as $\Delta t = 29$, $b = 1.8 \cdot 10^{-3}$, and the random forcing has standard deviation 0.002.

The prediction model uses a first guess for $\mathbf{q}(t)$ as $0.5\mathbf{q}_p(t)$ where $\mathbf{q}_p(t)$ are the true random forcing values, and the initial conditions are too-large as $1.5\mathbf{x}(0)$. Noisy observations $\mathbf{y}(t)$ are supposed to exist at the positions given in Fig. 10, and the measurement noise has a standard deviation of 0.001.

The field $\psi(t = 167\Delta t)$ as given by the true model is shown in Fig. 10, keeping in mind that apart from the time-mean $\psi$, the structure seen is the result of a particular set of random forcings.

*Aliasing*

In isolation, the observations will time-alias the field, if not taken at minimum intervals of 1/2 the shortest period present (here $4\Delta t$). A spatial-alias occurs if the separation between observations is less than 1/2 the shortest wavelength present (here





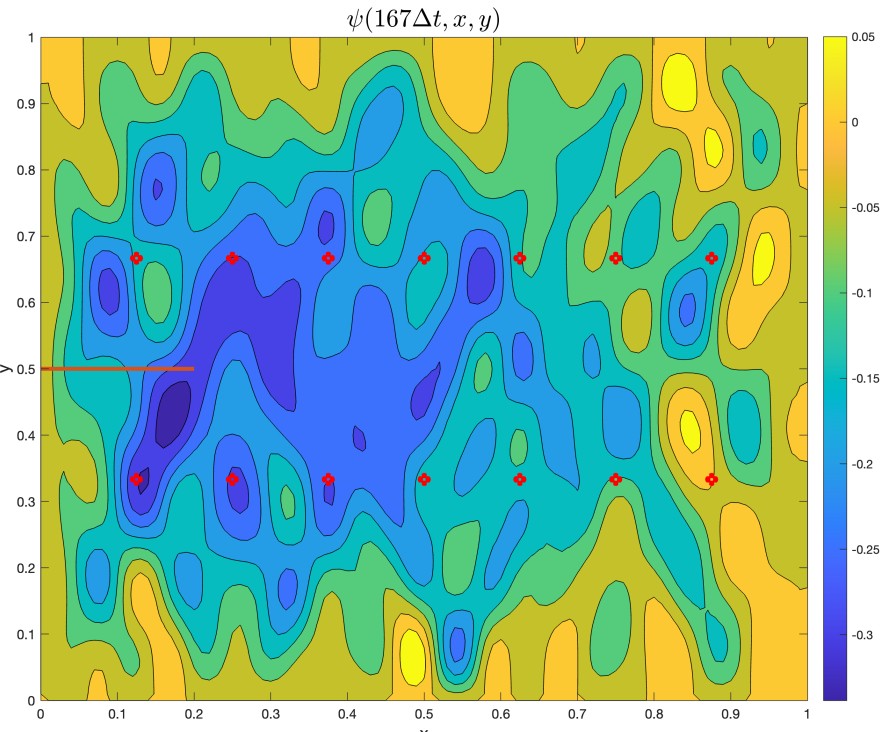

**Figure 10.** Stream function at $t = 167\Delta t$, with the normal modes superimposed on the time–independent Stommel solution. At later times, the mean flow becomes difficult to visually detect in the presence of the growing normal modes under the forcing. The markers indicate the locations of the observational data, and the horizontal line at $y = 0.5$ is the distance over which the boundary current transport is defined.

$\Delta y = 1/9$). Both these phenomena are present in what follows, but their impact is minimized by the presence of the time-evolution model.

## 4.2 Results: Kalman filter and RTS smoother

For the KF and RTS algorithms the model is run for $t_f = 2000$ timesteps with the above parameters.

### 4.2.1 Energy Estimates

A KF estimate is computed and the results stored. As in section 3.1.1, observations are introduced in two intervals, each with a different density of observations: initially data are introduced with $50\Delta t$ between them, and subsequently reduced to $25\Delta t$ between observations. Observations cease prior to $T_F$, mimicking a pure prediction interval following the observations.

The estimated values of the quadratic $\Phi(t)$ are shown in Fig. 11 for the true values, KF estimates, RTS smoother estimates, and the pure model prediction. The KF and prediction estimates agree until the first observation time, at which point a clear



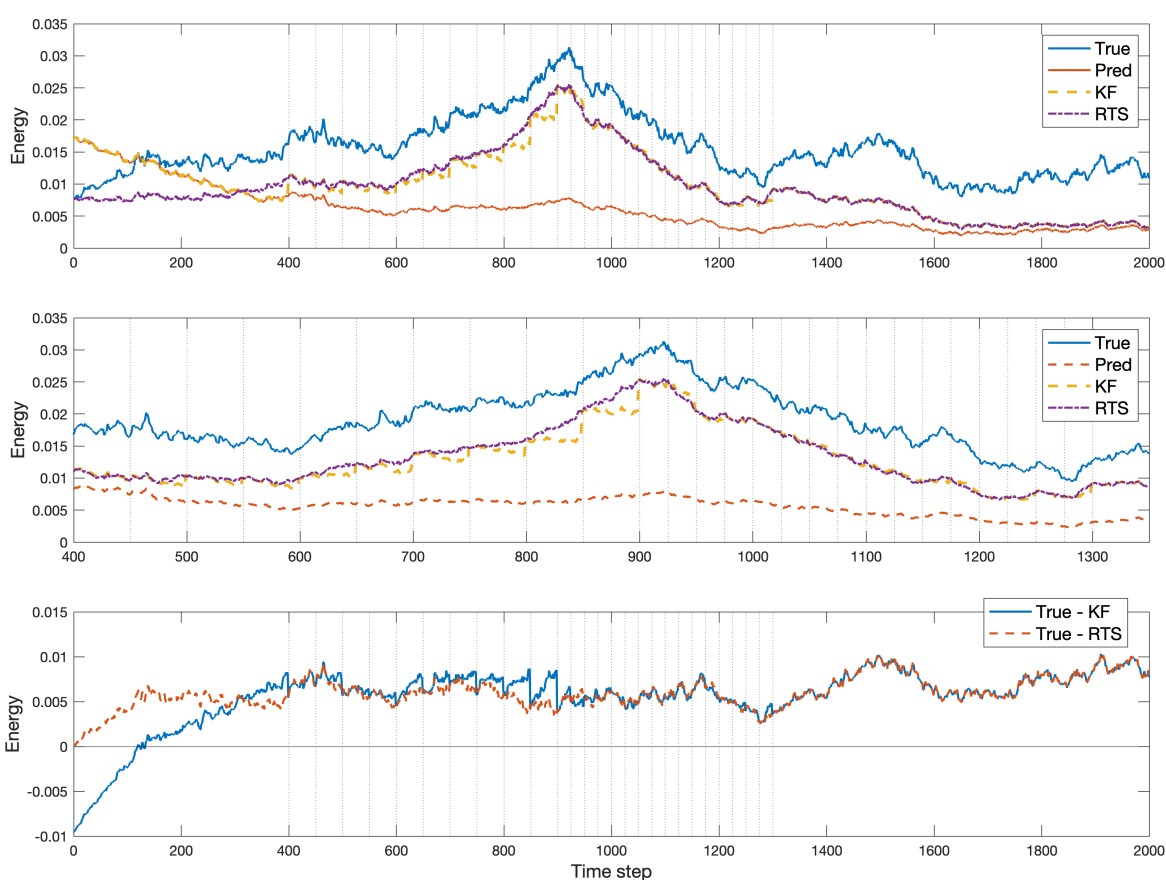

**Figure 11.** (a) The energy, i.e. $\Phi(t)$ defined by Eq. (31), computed from the non-dimensional state vector of the true model, prediction, KF, and the RTS reconstructions. In the KF the initial conditions are 20% too large, and the forcing is 50% too small. Full knowledge of **A** is assumed. (b) An expanded plot for timesteps $380 \leq t \leq 1325$ where observations were incorporated into the result. The RTS smoother produces an improved energy, namely one without discontinuities and marginally more accurate than does the KF. (c) Difference between the true values and from the KF (solid line), and between the and RTS smoother values (dashed line).



discontinuity is seen. As additional observations accumulate, $\Phi_{KF}(t)$ jumps by varying amounts depending upon the particulars of the observations and their noise. Over the entire observation interval the energy reconstructed by the KF remains low—a

systematic error owing to the sparse observations and null space of $\mathbf{E}$. Here the forcing amplitude overall dominates the effects of the incorrect initial conditions. Uncertainty estimates for energy would once-again come from summations of correlated $\chi^2$ variables of differing means. In the present case, important systematic errors are visible as the offsets between the curves in Fig. 11.

This system can theoretically be over-determined by letting the number of observations at time $t$ exceed the number of

unknowns—should the null space of $\mathbf{E}(t)$ then vanish. As expected, with 14 covarying observations, and 18 time-varying unknown $x_i(t)$, rank 12 time-independent $\mathbf{E}(t) = \mathbf{E}$ has a nullspace, and thus energy in the true field is missed even if the observations were perfect. As is well-known in inverse methods, the smaller eigenvalues of $\mathbf{E}$ and their corresponding eigenvectors are most susceptible to noise biases. The solution nullspace of this particular $\mathbf{E}(t)$ is found from the solution eigenvectors of the singular value decomposition, $\mathbf{U}\Lambda\mathbf{V}^T = \mathbf{E}$. Solution resolution matrix at rank $K = 12$, $\mathbf{V}_K\mathbf{V}_K^T$, is shown in Fig. 12.

Thus the observations carry no information about modes (as ordered) 3, 6, 9, 12, 15, and 18. In a real situation, if control over positioning of the observations was possible, this result could sensibly be modified and/or a strengthening of the weaker singular values could be achieved. Knowledge of the nullspace structure is important in the interpretation of results.

A more general discussion of nullspaces involves that of the weighted $\mathbf{P}(\tau, -)\mathbf{E}^T$ appearing in the Kalman gain (Eq. 5). If $\mathbf{P}(\tau, -)^{1/2}$ is the Cholesky factor of $\mathbf{P}(\tau, -)$ (Wunsch (2006), page 56), then $\mathbf{EP}(\tau, -)^{1/2}$ is the conventional column-

weighting of $\mathbf{E}$ at time $\tau$, and the resolution analysis would be applied to that combination. A diagonal $\mathbf{A}$ does not distribute information from any covariance amongst the elements $x_j(\tau)$ and which would be carried in $\mathbf{P}(\tau, -)$.

Turning now to the RTS smoother, Fig. 11 shows that the energy in the smoothed solution, $\Phi_{RTS}(t)$, is continuous (up to the model time-stepping changes), but briefly exceeds the true energy prior to the appearance of the first observation. The only information available to the prediction prior to the observational interval lies in the initial conditions, which were given

incorrect values leading to an initial uncertainty. Estimated unknown elements $\mathbf{u}(t)$, of $\mathbf{q}(t)$ in this interval also have a large variance.

One element through time of the estimated control vector, $\tilde{\mathbf{u}}(t)$, and its standard error are shown in Fig. 13. The complex result of the insertion of data is apparent. As with the KF, discussion of any systematic errors has to take place outside of the formalities leading to the smoothed solution. This RTS solution does conserve $\Phi(t)$, as well as other properties (circulation).

**4.2.2 Western Boundary Current Estimates**

Consider now determination of $\tilde{T}_{WBC}(t)$, the north-south transport across latitude $y_0$ at each time-step, whose true value is constant. $T_{WBC}(t)$ is computed from the velocity or stream function as,

$$T_{WBC}(t) = \psi(t, 0, y_0) - \psi(t, x_0, y_0), \tag{33}$$

the stream-function difference between a longitude pair, $x = 0$, $x_0$. From the boundary condition, $\psi(t, 0, y_0) = 0$, identically.

The horizontal line segment in Fig. 10 indicates the location of the zonal section for the experiment at $y_0 = 0.5$, extending from





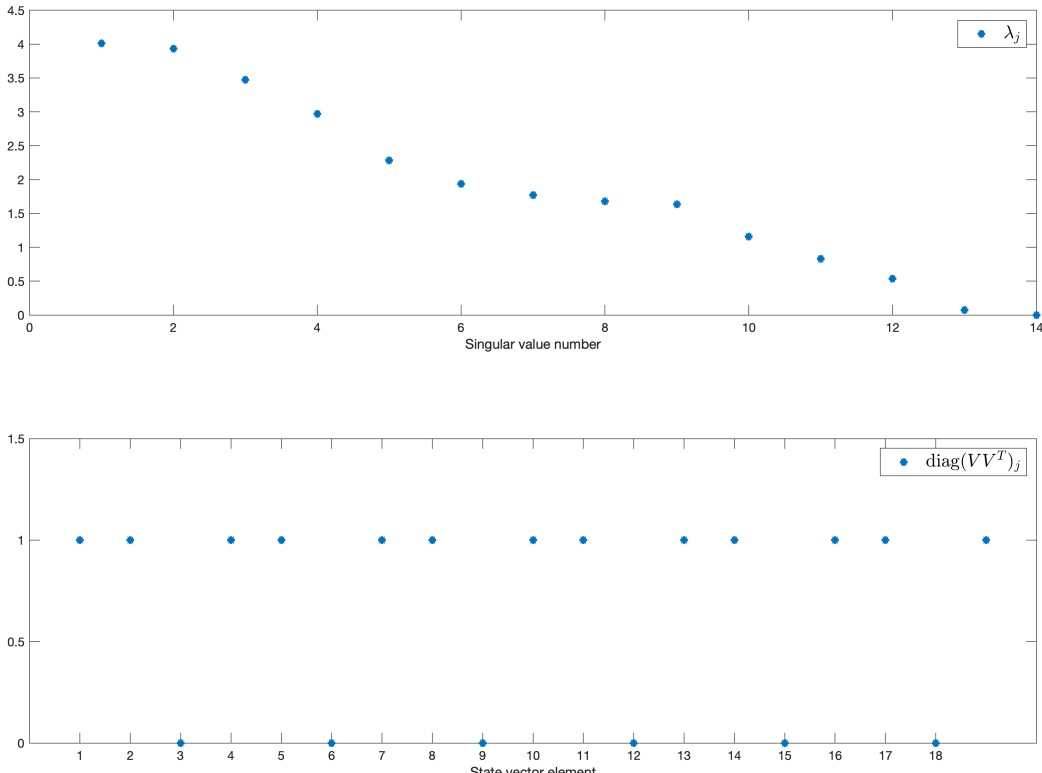

**Figure 12.** (a) Eigenvalues of the first 14 singular vectors of **E**. Rank is 12 with 14 observational positions. (b) Diagonal elements of the rank 12 solution resolution matrix, showing lack of information for several of the modes. A value of 1 means that the mode is fully resolved by the observations. All variables are non-dimensional.

$x = 0$ to $x_0 = 0.2$. In the present context, five different values of $T_{WBC}(t)$ are relevant: (a) the true, steady, time-invariant value, computed from the Stommel solution; (b) the true apparent value including mode contributions from Eq. (27); (c) the estimated value from the prediction model; (d) the estimate from the KF; and (e) the estimate from the RTS smoother. Fig.14a displays the transport computed from the full version of Eq. 33, alongside the transport computed from the KF estimate. Panel (b) displays the same variables for the RTS estimate. Values here are dominated by the variability induced by the normal modes, leading to a random walk. Note that the result can depend sensitively on positions $x_0, y_0$, and the particular spatial structure of any given normal mode.

In the KF reconstruction (Fig. 14a), observations move the WBC transport values closer to the steady solution, seen via the jump at $t = 400$, but remain noisy. Transport value uncertainties are derived from the **P** of the state vector using Eq. (A1) and shown in Fig. 14. Within the observation interval the estimates are indistinguishable from the true value, but still have a wide



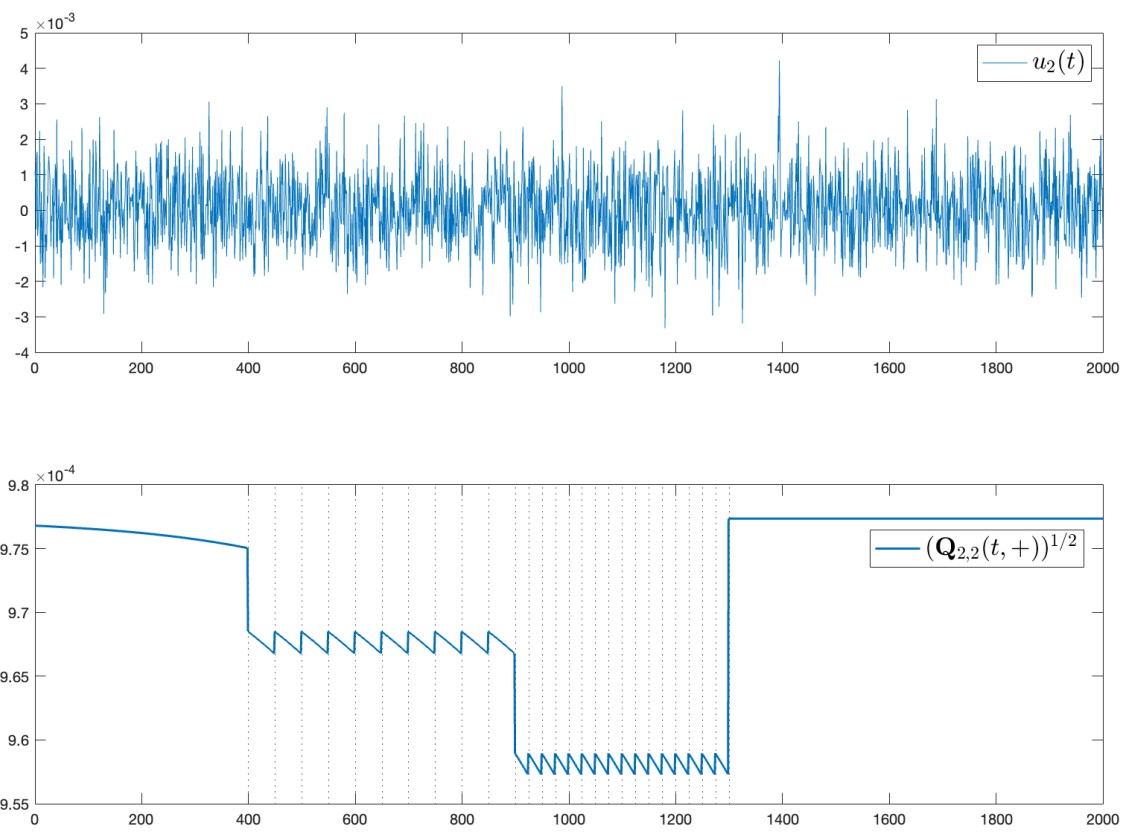

**Figure 13.** (a) One element, $u_2(t)$, of the control vector correction estimate and (b) its standard error through time, showing the drop towards zero at the data time, and the slow increase towards a higher value when no data are available.

uncertainty with time scales present both from the natural variability and the regular injection times of the data. The magnitude of the uncertainty, during the observation intervals, is still roughly 10% of the magnitude of the KF estimate.

Fig. 14b shows the behavior of the estimate of $T_{WBC}(t)$ after the RTS smoother has been applied. Most noticeably, the discontinuity that occurred at the onset of the observations has been removed.

A test of the negative hypothesis that the transport computed from the RTS smoother was indistinguishable from a steady value is based upon an analysis using the uncertainty (not shown).

The very large uncertainty prior to the onset of data, even with use of a smoothing algorithm, is a central reason that the state estimate produced by the Estimating the Circulation and Climate of the Ocean (ECCO) project (e.g. Fukumori et al. (2018)) is confined to the interval following 1992 when the data become far denser than before. Estimates prior to a dense data interval
depend greatly upon the time durations built into the system, which in the present case are limited by the longest normal mode



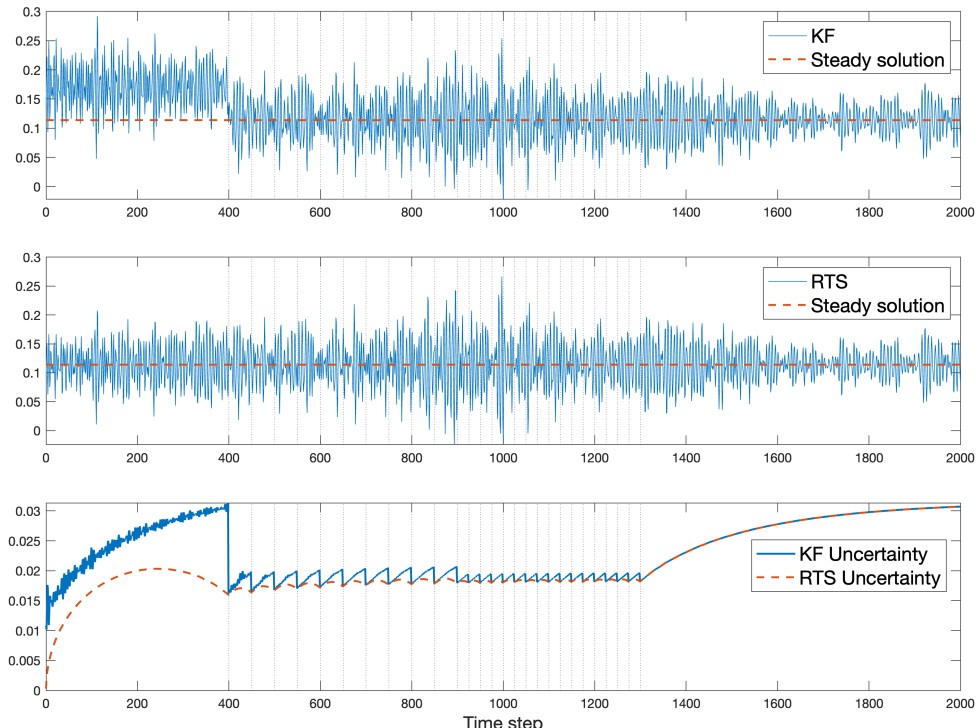

**Figure 14.** (a) Estimated non-dimensional western boundary current transport from the Kalman filter (solid line) and the steady western boundary current from the Stommel solution (dashed line). (b) Same as (a) except now the smoothed $T_{WBC}$ from the RTS smoother next to the steady solution. Vertical dashed lines indicate timesteps where data was available (c) Uncertainty in the $T_{WBC}$ predictions over time, computed from the operators $\mathbf{P}(t)$ and $\mathbf{P}(t,+)$.

period. The real ocean does include some very long memory (Wunsch and Heimbach (2008)), but the estimation skill will depend directly on the specific physical variables of concern. (ECCO estimates are based upon a different algorithm using iterative least-squares and Lagrange multipliers (Stammer et al. (2002).) For a linear system, those results are identical to those using a sequential smoother; see Fukumori et al. (2018).)

Some understanding of the impact via the smoother of later observations on KF time estimates can be found from the operator $\mathbf{L}(t)$. Fig. 15 shows the norm of the operator $\mathbf{L}$ (Eq. 20) controlling the correction to earlier state estimates, along with the time dependence of one of its diagonal elements. As always, the temporal structure of $\mathbf{L}(t)$ depends directly upon the time constants embedded in $\mathbf{A}$, and the compositions of $\mathbf{P}(t), \mathbf{P}(t+\Delta t,-)$. In turn these latter are determined by any earlier information, including initial conditions, as well as the magnitudes and distributions of later forcing and data accuracies.

Generalizations are not easy.





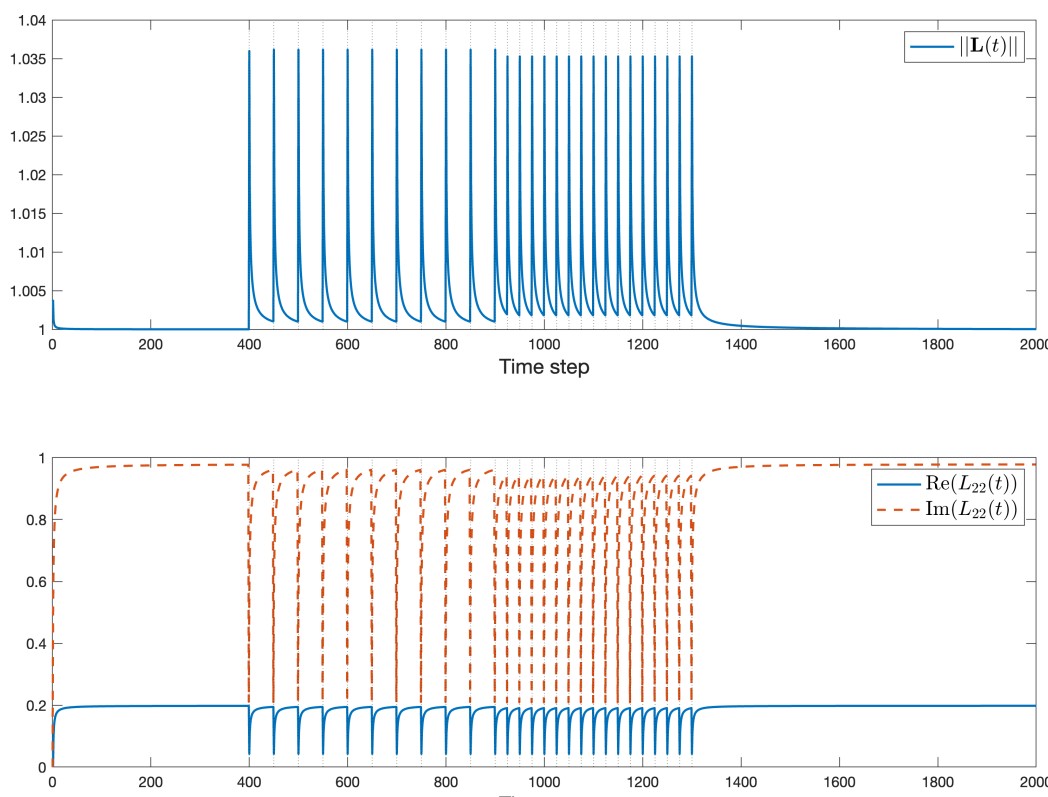

**Figure 15.** (a) Norm of the operator $\mathbf{L}$ controlling the backwards in time state estimate (see (20), RTS smoother for the full equation) (b) The real (solid line) and imaginary (dashed line) components of $L_{22}(t)$.

The norm of the gain matrix $\mathbf{M}(t)$, used for computation of the control vector Eq. (A4), provides a measure of its importance relative to the prior estimate, and is displayed in Fig. 16. Here the dependence is directly upon the a priori known control variance $\mathbf{Q}(t)$, the data distributions, and $\mathbf{P}(t + \Delta t, -)$. The limiting cases discussed above for the state vector also provide insights here.

### 4.2.3 Spectra

Computation of the spectral estimates of the various estimates of any state vector element or combination is straightforward and the $z-$transforms in the Appendix provide an analytic approach. What is not so straightforward is the interpretation of the result in this non-statistically stationary system. Care must be taken to account for the non-stationarity.





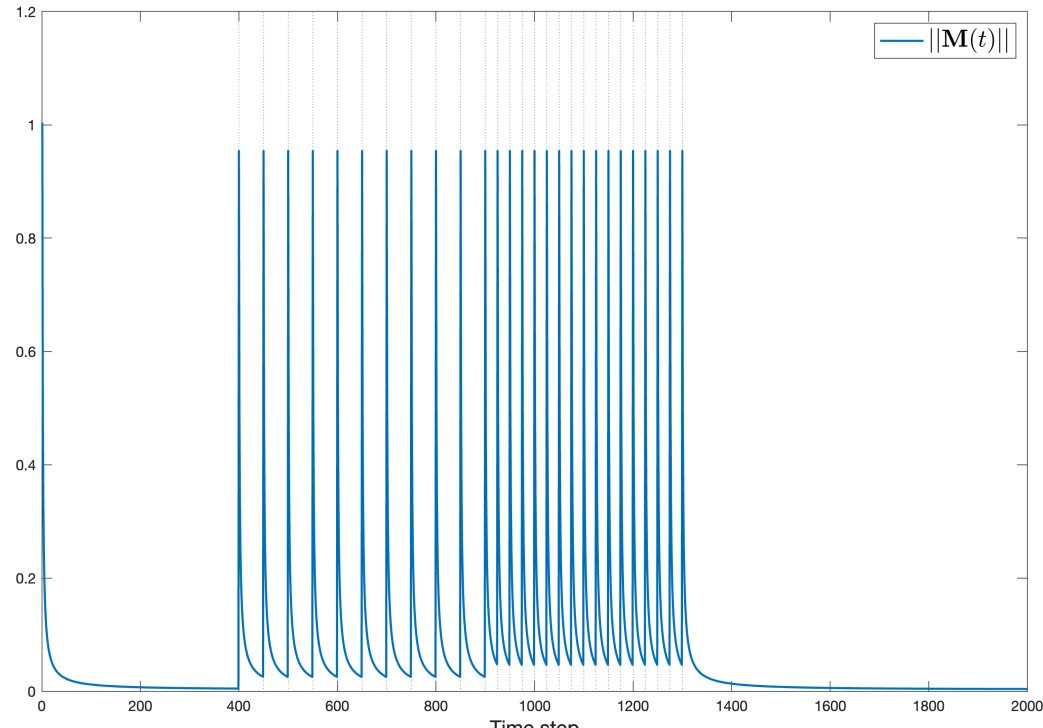

**Figure 16.** Norm of the gain matrix $\mathbf{M}$ through time and which determines the magnitude and persistence of inferred changes in the control variables $\mathbf{u}(t)$.

## 5 Discussion

In sequential estimation methods, the behavior of dynamical system invariants and conservation laws, including energy or circulation or scalar inventories, or derived ones such as a thermocline depth, depend, as shown using toy models, upon a number of parameters. These parameters include the time scales embedded in the dynamical system, the temporal distribution of the data relative to the embedded time-scales, the accuracies of initial conditions, boundary conditions, sources and sinks, and data, as well as the accuracy of the governing time evolution model. Errors in any of these parameters can lead to physically

significant errors in estimates of the state and control vectors and any quantity derived from them.

Estimates depend directly upon the accuracies of the assumed and calculated uncertainties in all of the elements making up the estimation system. Impacts of data insertions can range from very short time intervals to those extending far into the future. Because of model/data interplay, the only easy generalization is that the user must check the accuracies of all of these elements, including the appearance of systematic errors in any of them (e.g., Dee (2005)), or of periodicities arising solely

from data distributions. When feasible, a strong clue to the presence of systematic errors in energies, as one example, lies in



determining the nullspace of the observation matrices coupled with the structure of the state evolution matrices, $\mathbf{A}$. Analogous examples have been computed for advection-diffusion systems (not shown, but see Marchal and Zhao (2021)) with results concerning, in one example, estimates of fixed total tracer inventories.

In the Rossby wave example, reconstructions of the constant western boundary current transport improved from that of the KF, by using *future* data and the Rauch-Tung-Striebel (RTS) smoother. Estimates nonetheless can still contain large uncertainties—quantifiable from the accompanying algorithmic equations. The RTS algorithm is only one choice from several approaches to the finite interval estimation problem. Alternatives include the least-squares/Lagrange multiplier approach of the ECCO project, which in the linear case can be demonstrated to produce identical results.

All possible error sources have not been explored. In particular, only linear systems were analyzed, assuming exact knowl-
edge of the state transition matrix, $\mathbf{A}$, and the data distribution matrix, $\mathbf{B}$. Notation was simplified by using only time-independent versions of them. Nonlinear problems arising from errors in $\mathbf{A}$, $\mathbf{B}$ will be described elsewhere. Other interesting nonlinear problem include those where the observations are not independent of the state vector, or the observations—such as a speed—are nonlinear in the state vector.

Unless, as in weather prediction, short-term predictions are almost immediately testable by comparison with the observed
outcome, physical insights into the system behavior are essential, along with an understanding of the structure of the imputed statistical relationships. As a considerable literature cited above has made clear, the inference of trends in properties and understanding of the physics (or biology or chemistry) in the presence of time-evolving observation systems requires particular attention. At a minimum, one should test any such system against the behavior of a known result—for example, treating a GCM as "truth" and then running the smoothing algorithms to test whether that truth is forthcoming. This approach guarantees
exact dynamical and kinematic consistency of the state estimate (Stammer et al. (2016), Wunsch and Heimbach (2007)) a key requirement when seeking physical understanding of the results.

## Appendix A:  Notation and Equations

*Kalman Filter*

The model state transition equation is that in Eq. (1) and the weighted averaging equation is Eq. (4) with the gain matrix
$\mathbf{K}(t)$ defined in Eq. (5). Time evolution of the covariance matrix of $\tilde{\mathbf{x}}(t)$ is governed by

$$\mathbf{P}(t,-) = \left\langle (\tilde{\mathbf{x}}(t,-) - \mathbf{x}(t))(\tilde{\mathbf{x}}(t,-) - \mathbf{x}(t))^T \right\rangle \tag{A1}$$
$$= \mathbf{A}(t)\mathbf{P}(t-\Delta t)\mathbf{A}(t)^T + \mathbf{\Gamma}(t-\Delta t)\mathbf{Q}(t-\Delta t)\mathbf{\Gamma}(t-\Delta t)^T,$$

and

$$\mathbf{P}(t) = \mathbf{P}(t,-) - \mathbf{K}(t)\mathbf{E}(t)\mathbf{P}(t,-) \tag{A2}$$
$$= \mathbf{P}(t,-) - \mathbf{P}(t,-)\mathbf{E}(t)^T \left[\mathbf{E}(t)\mathbf{P}(t,-)\mathbf{E}(t)^T + \mathbf{R}(t)\right]^{-1}\mathbf{E}(t)\mathbf{P}(t,-),$$



$\mathbf{E}, \mathbf{R}$ are defined in the text. The matrix symbol $\mathbf{\Gamma}$ is introduced for a situation in which the control distribution over the state differs from that in $\mathbf{B}$. Otherwise they are identical. Because $\mathbf{P}$ is square of the state vector length, calculating it is normally the major computational burden in the use of a Kalman filter.

Under some circumstances where a system including observation injection reaches a steady state, the time-index may be omitted in both the KF and the RTS smoother. Time independence is commonly assumed when the rigorous formulation for the KF is replaced by an ad hoc constant gain matrix $\mathbf{K}$.

*RTS Smoother*

In addition to Eqs. (19), (20), the Rauch-Tung-Striebel smoother estimates

$$\tilde{\mathbf{u}}(t,+) = \mathbf{M}(t+\Delta t)\left[\tilde{\mathbf{x}}(t+\Delta t,+) - \tilde{\mathbf{x}}(t+\Delta t,-)\right] \tag{A3}$$

$$\mathbf{M}(t+\Delta t) = \mathbf{Q}(t)\mathbf{\Gamma}(t)^T\mathbf{P}(t+\Delta t,-)^{-1}, \tag{A4}$$

for the updated control $\mathbf{u}(t)$. $\mathbf{Q}(t)$ is the assumed covariance of $\mathbf{u}(t)$ (the uncertainty in $\mathbf{q}(t)$) and $\mathbf{\Gamma}$ is again often equal to $\mathbf{B}$. Then the corresponding uncertainties of the smoothed estimates are,

$$\mathbf{P}(t,+) = \mathbf{P}(t) + \mathbf{L}(t+\Delta t)\left[\mathbf{P}(t+\Delta t,+) - \mathbf{P}(t+\Delta t,-)\right]\mathbf{L}(t+\Delta t)^T, \tag{A5a}$$

$$\mathbf{P}_u(t,+) = \mathbf{Q}(t,+) = \mathbf{Q}(t) + \mathbf{M}(t+\Delta t)\left[\mathbf{P}(t+\Delta t,+) - \mathbf{P}(t+\Delta t,-)\right]\mathbf{M}(t+\Delta t)^T, \tag{A5b}$$

One can gain insight into this filter/smoother machinery by considering its operation on a scalar state vector with scalar observations.

## Appendix: Green Function Analysis of Estimates

*KF response*

The impact at other times of having data at time $t$ can lend important physical insight into the sequential analyses. Define an innovation *matrix,*

$$\mathbf{D}_\delta(t,j) = \mathbf{y}(t) - \mathbf{E}(t)\mathbf{x}(t) = \delta_{t,\tau}\boldsymbol{\delta}_{ij} \tag{A6}$$

that is, $\mathbf{D}_\delta$ is a matrix of Kronecker deltas representing the difference $D_{ij}(\tau) = \delta_{t,\tau}\boldsymbol{\delta}_{ij} = y_j(\tau) - \sum_r E_{ir}(\tau)x_r(\tau)$. Solutions to the innovation equation are the columns of the Green function matrix,

$$\mathbf{G}(t) = \mathbf{A}\mathbf{G}(t-\Delta t) + \mathbf{K}\mathbf{D}_\delta(t), \ t = m\Delta t. \tag{A7}$$

$\mathbf{K}$, now fixed in time, is sought as an indication of a delta impulse effects of observations on the prediction model at time $\tau$. Define the scalar complex variable,

$$z = \exp(-i2\pi s\Delta t), -1/2\Delta t \le s \le 1/2\Delta t. \tag{A8}$$





where $s$ is the frequency. Then the discrete Fourier transform of Eq. (A7) (the $z-$transform—a matrix polynomial in $z$) is,

$$\hat{\mathbf{G}}(z) = (\mathbf{I} - z\mathbf{A})^{-1}\mathbf{K}\hat{\mathbf{D}}_\delta(z). \tag{A9}$$

The norm of the variable $(\mathbf{I} - z\mathbf{A})^{-1}$ defines the "resolvent" of $\mathbf{A}$ in the full complex plane (see Trefethen and Embree, 2005), but here, only $|z| = 1$, is of direct interest, that is, only on the unit circle. The full complex plane carries information about the behavior of $\mathbf{A}$, including stability.

Here $\hat{\mathbf{D}}(z) = \mathbf{I}z^\tau$ and,

$$\hat{\mathbf{G}}(z) = (\mathbf{I} - z\mathbf{A})^{-1}\mathbf{K}z^\tau \tag{A10}$$

If a suitably defined norm of $\mathbf{A}$ is less than 1,

$$\hat{\mathbf{G}}(z) = (\mathbf{I} - z\mathbf{A})^{-1}\mathbf{K}z^\tau \approx \left(z^\tau\mathbf{I} + z^{\tau+1}\mathbf{A} + z^{\tau+2}\mathbf{A}^2 + z^{\tau+3}\mathbf{A} + ...\right)\mathbf{K} \tag{A11}$$

and the solution matrix in time is the causal vector sequence (no disturbance before $t = \tau$) of columns of

$$\mathbf{G}(t) = 0, t < \tau \tag{A12}$$
$$= \mathbf{A}^m\mathbf{K}(\tau), t = \tau + m\Delta t$$

$m = 0, 1, 2, ...$

$\mathbf{G}$ can be obtained without the $z-$transform, but the frequency content of these results is of interest.

*Green Function of Smoother Innovation*

As with the innovation equation for filtering, Eq. (19) introduces a disturbance into the previous estimate, $\tilde{\mathbf{x}}(t)$, in which the structure of $\mathbf{L}(t)$ determines the magnitude and time scales of observational disturbances propagated *backwards* in time. It

provides direct insight in the extent to which later measurements influence earlier ones. As an example, suppose that the KF has been run to time $t = T_F$ so that $\tilde{\mathbf{x}}(T_f, +) = \tilde{\mathbf{x}}(T_f)$, which is the only measurement. Let the innovation, $\tilde{\mathbf{x}}(T_f, +) - \tilde{\mathbf{x}}(T_f, -)$, be a matrix of $\delta$ functions in separate columns,

$$\mathbf{D} = \delta(t - T_f)\mathbf{I}_N \tag{A13}$$

then a backwards-in-time matrix Green function is,

$$\mathbf{G}(t) = \mathbf{L}(t)...\mathbf{L}(T_f - \Delta t)\mathbf{L}(T_f) \tag{A14}$$

The various time-scales embedded in $\mathbf{L}$ depend upon those in $\mathbf{A}, \mathbf{P}(t, -), \mathbf{P}(t)$ and with many observations including those of the observation intervals, and any structure in the observational noise. Similarly, the control modification will be determined by $\mathbf{P}(t + \Delta t, -)^{-1}$ if $\mathbf{Q}(t)\mathbf{\Gamma}(t)^T$ are constant in time.

*Author contributions.* Calculations were done primarily by Williamson. Preliminary calculations and the initial writing were by Wunsch.
Heimbach, advised, read the ms. and checked accuracies



*Competing interests.* None

*Acknowledgements.* Supported from the NASA/UT Austin/JPL ECCO Projects. Work by CW done at home originally during the Trump-Covid Apocalypse period. We would like to thank the two anonymous reviewers for providing detailed, helpful comments on the first copy of this manuscript. Detlef Stammer provided many useful suggestions for an earlier version of the manuscript.

*Code availability.* Matlab codes used here are available directly from SW Github page: https://github.com/swilliamson7/data_assimilation_project.



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
