# Peer review of "Potential Artifacts in Conservation Laws and Invariants Inferred from Sequential State Estimation"

_EGUsphere, 2023_

## Referee Comment (RC1)

This manuscript is a revised version of a similar manuscript by the first author which I reviewed some time ago. As a result, some, not all, of the points made in this review are borrowed from my first review.

The authors explore potential artifacts in the estimates of physical invariants which are obtained from the quantitative combination of time series data and dynamical models using sequential estimation procedures (Kalman filter and related smoother). Two physical systems are being studied: a system of three coupled oscillators and the wind-driven flow of a uniform-density fluid in a closed square basin. Emphasis is placed on the determination of "trends" of quantities of oceanographic/climatic relevance, such as mechanical energy and western boundary currents. It is emphasized, among other results, that (i) entry of data in the filter leads to violation of conservation laws while smoothing restores conservation rules; and (ii) a robust identification of trends in reanalysis estimates requires a complete understanding of both models and observations.

The determination of trends in oceanographic time series such as produced by basin-scale monitoring programs (e.g., RAPID and OSNAP) is a subject of growing interest. The study of trends in simplified physical systems such as reported here is essential for developing a better understanding of the challenges associated with the determination of trends in more realistic situations. The approach applied in the present manuscript is inspiring. Although the models employed are idealized descriptions, they permit a discussion of the some of the major issues, such as those associated with the nature of the data, their uncertainties, as well as their spatial and temporal distributions, which are likely to occur in more realistic situations. Overall, I think this manuscript should be a significant contribution, one that will hopefully invite researchers to interpret cautiously apparent trends or similar features in reanalysis estimates.

Reading through the manuscript, it appears that most, but not all, of the comments of my first review have been considered. Major comments are listed below, followed by a list of specific points. I hope these will help the authors to improve their very interesting work.

MAJOR COMMENTS

1) I am intrigued by the statement in the Abstract that a finite interval smoother restores conservation rules which are violated by the Kalman filter. Much like the Kalman filter (KF) estimates, Rauch-Tung-Striebel (RTS) smoother estimates are weighted least-squares estimates, with the weighting provided by data and model error covariances (see Bryson and Ho, 1975; in particular their Problem 3 on p. 395). As the authors are well aware, the basic difference between the KF estimates and the RTS estimates relies in the time span of the data being used. KF estimates are based on past and present data, whereas RTS estimates, which have the appearance of a correction to KF estimates (Eq. 19), are based on past, present, and future data. In fact, an optimum linear smoother such as the RTS smoother can be interpreted as a combination of two optimum linear filters, one which works forward over the data interval and the other which runs backward over the interval (Fraser and Potter 1969). From these considerations, I would expect the RTS smoother to share the same limitation as the KF filter, which is the introduction, when data are entered, of a violation of conservations laws as implemented in the dynamical model.

The above considerations, however, are purely qualitative and might be misleading. The statement in the Abstract is justified in the text by the fact that the system forcing q(t) is modified during smoothing (p. 17). While it is true that q(t) is modified during smoothing, it is unclear to me that the state evolution forced only by the modified q(t) and which would thus satisfy conservation laws is the same as the state evolution computed from smoothing. I would recommend that the authors illustrate in the manuscript a numerical example that would support their statement.

2) The manuscript reports results from a large number of filtering and smoothing experiments. To guide the reader through them, it would be useful to label the different experiments in the text, and to add a table where the design of the different experiments is described (the authors made a welcome move in this direction through their Table 1). For convenience, the first reported KF experiment for the coupled oscillators could be labelled as KF1-O-n, where n is a number or (perhaps better) an evocative character string. One table would list the experiments for the system of coupled oscillators, and another table would list the experiments for the wind-driven flow.

3) The choice $L = a$ ($a$ = the Earth radius) to scale the potential vorticity equation (21) is not consistent with the beta-plane approximation. A choice of $L$ consistent with the beta-plane approximation should be assumed (e.g., Pedlosky 1987).

4) I think both terms on the right-hand side of equation (29) (boundary-layer solution of Stommel model) must be divided by beta'.

5) The Discussion section is very short, and the manuscript lacks a Conclusion or Summary section. I would suggest to extend the text, so that an explicit concluding section appears at the end of the manuscript.

SPECIFIC POINTS

L. 5: "… oscillator system and …"

L. 12: "… the climate system as represented in a reanalysis. As a result, some simple …"

L. 21: "… As in geophysical fluid dynamics, two …"

L. 25: "… and models, and both …"

L. 61: "… transport of a western boundary current."

L. 99: "… mean error), respectively. The operator $P(t,-)$ …". Also here and everywhere: I would suggest labeling $P(t)$, $P(t,-)$, $P(t,+)$, and similar matrices, not as operators but as error covariance matrices or uncertainties.

L. 100: "… evolve …"

L. 103: "… both $y(t+\Delta t)$ and $E(t+\Delta t)$ vanish, …" ($y$ is a vector while $E$ is (generally) a matrix, so they could not be set to the same value).

L. 107: "… and $R(\tau)$."

L. 115: "3. Example 1: System of Mass-Spring Oscillators"

L. 116: "… system of mass-spring oscillators, following …"

L. 151: "Equation (1) is considered at …" (Eq. 1 is a difference, not differential equation, and so is already discretized in some sense).

L. 176: "…, data, etc."

L. 188: "Added white noise in the data …". Please make sure to use different symbols for the noise in the data and in the model equations.

Fig. 3, caption: "… Energy varies with the purely random process $\varepsilon(t)$ as well as …"

L. 205: Please elaborate.

Figure 4: the ordinate of the bottom panel is "velocity", not "displacement". In the caption, perhaps write "… (c) Estimated velocity of the first …"

L. 264-265: "… observations (see Appendix A; Eq. A3). Suppose …"

L. 305: "… horizontal dimension L. Here $\beta = df/dy$ is the variation of the Coriolis parameter, f, with the latitude coordinate, y. This problem …".

L. 313: Drop " $q = q_0 q'$ " and write " $\psi = (a^2 f) \psi'$ "

L. 333: "…the coefficients $c_p$ do not strictly …"

L. 336: "… by addition to $\psi$ of …"

L. 339: "… Rayleigh friction coefficient."

L. 343: "… wind-stress curl."

L. 347-348, "… diagonal elements diag(…) … NM+1.": Please rephrase.

L. 350: "… M diagonal elements equal to …"

L. 366: "… generate the true fields."

L. 368: "… the random forcing $q_j$ has …"

L. 377: "… is more than …"

L. 385: "… Observations cease posterior to $T_F$, …"

Fig. 11, caption: "… (c) Difference between the true values and the KF values (solid line), and difference between the true values and the RTS smoother values (dashed line). The vertical lines show …"

L. 408, "… briefly exceeds the true energy prior to the appearance of the first observations": I do not see this in figure 11.

L. 410, "… Estimated unknown elements u(t) …": Please remind the reader what u(t) represents.

L. 422: "… (b) the true time-dependent value …"

L. 424: do you mean "… full solution of Eq. (28)"?

Fig. 13, caption, "… slow increase towards a higher value when no data are available.": This is not apparent in figure 13b.

L. 435: "A test of the null hypothesis that …"

L. 444: "… smoother (see Fukumori et al. 2018)."

Fig. 15, caption: "… equation), (b) …"

L. 482, "… observations are not independent of the state vector …": Please drop or rephrase.

L. 490: "… Wunsch and Heimbach (2007)), a key …"

Eq. (A1): "… = A(t-$\Delta$t) P(t-$\Delta$t) A(t-$\Delta$t)$^T$ + …"

L. 501-502: I do not see the rationale for introducing the two distribution matrices, $\Gamma$ and B. It seems to me that $\Gamma$ and B are always the same matrix (e.g., Bryson and Ho 1975).

L. 517: "Appendix B: …"

L. 520: "… innovation vector,"

L. 523, "innovation equation": Please identify or report this equation in the manuscript.

L. 551-552, "… depend upon … and with many observations including those …": Please rephrase.

---

## Author Comment (AC1)

**Response to Reviewer #1**

We would like to thank reviewer #1 for their thoughtful comments on the new revision of the manuscript. We begin by responding to the major comments given:

1. The first major comment discusses whether the outcome of the RTS smoother estimate truly restores conservation rules that were violated by a Kalman filter estimate. The reference to Bryson and Ho (1975),and in particular Problem 3 on page 395, was helpful. We outline below how this problem proves that the RTS estimate restores violations in conservation laws introduced by the KF.

   Bryson & Ho (1975; in the following BH75) lay out the solution to the KS problem in three steps (sections 13.1–13.4), showing that updates to the initial conditions and external forcings, obtained recursively, are then used as inputs to the free-running model:
   (1) Considering a single-state transition, BH75 show how a correction to the state at time $t = 0$ (initial condition) and external forcing are obtained using observations at time $t = 1$.
   (ii) Extending this to a multi-stage process, the initial conditions subject to future observations and forcing updates at times $t = i$ are obtained recursively (eqns. 13.2.1–13.2.4), but requiring non-trivial computations in practice (rendering an exact implementation of the KS difficult for complex applications).
   (iii) The general smoothing process of a nonlinear dynamical system is sketched in BH75, section 13.4. Again, upon computing smoother updates to the forcing and initial condition, the state evolution is obtained via the direct integration of the forward model, thus fulfilling all conservation or invariance principles imposed by the underlying governing equations.

   The issue is briefly addressed in the manuscript in lines 302–307, and a reference to Bryson and Ho has been added.

2. The second major comment was in regards to labelling and outlining the different experiments described in the text. As the reviewer notes, we made an effort towards this with Table 1, and do feel that this does a good job of describing the parameters chosen for each experiment conducted with the mass-spring oscillator toy problem. Labelling the experiments with designated names is useful is certain circumstances, but here the experiment discussions are largely self contained, so references to them outside of their respective sections can be best done via a section reference itself.

   Regarding adding a table for the Rossby wave experiments, this does not seem necessary as only one experiment setup is actually considered. We attempted to clearly outline the setup for the truth and prediction in Section 4.1.1., before beginning discussions on energy estimates and WBC estimates. Once the prediction setup is known, alongside the noise that was added to the data points and the location of data, the KF and RTS assumptions follow. We run the KF to find the prediction $\tilde{x}(t)$, and subsequently the RTS smoother $\tilde{x}(t, +)$, both only once. From these modal estimates we compute we find the energy and WBC estimates.

3. Major comment 3 was with regards to our choice of $L = a$ where $a$ is the radius of the Earth. We chose $L = a$ because the width of the Pacific is the same order of magnitude as the radius of the Earth. It is coincidental but in the spirit of a toy model, it does eliminate one more non-essential parameter from the system. (It does not change the physical character of the solution.)

4. Major comment 4 was in regards to Eq. (29), the approximate solution to the Stommel equation. Namely, it was suggested to divide all terms on the RHS by $\beta'$. This solution is taken from Pedlosky (1965) (Eq. 5.11), and we think it is correct as is.

5. The final major comment was in regards to the length of the discussion section, and that the manuscript lacks a conclusion or summary section. We have relabelled the Discussion section to be Discussion and Conclusions, as we intended for this section to conclude the manuscript. We have also expanded the end of the Conclusions section slightly in response to comments by Reviewer #2.

With regards to the rigorous, "specific points" about typos and suggested rewrites, all have been reviewed and applied where necessary in our effort improve the manuscript. These were very helpful and greatly appreciated. Below we respond to several of them directly. We have reproduced the reviewer's comments in *italic*:

*L. 205: Please elaborate.*

We have added the following sentence:
"The uncertainty in the predicted displacement is small once new data are inserted via the analysis increment, but it quickly grows as the model is integrated beyond the time of analysis."

*Fig. 13, caption, "... slow increase towards a higher value when no data are available.": This is not apparent in figure 13b.*

To clarify our statement, we have added the following sentence in the caption:
"(we note here that $\mathbf{Q}(t, +)$ is computed from $t = T_f$ backwards to $t = 0$)."

*L. 482, "... observations are not independent of the state vector...": Please drop or rephrase.*

We have rephrased as follows:
"... observations are derived quantities of the state vector..."

*L. 501-502: I do not see the rationale for introducing the two distribution matrices, $\Gamma$ and $\mathbf{B}$. It seems to me that $\Gamma$ and $\mathbf{B}$ are always the same matrix (e.g., Bryson and Ho 1975).*

In general, $\mathbf{B}$ distributes the known part of forcing, whereas $\Gamma$ distributes the unknown part of the forcing or controls, as in the Rossby wave example with $\mathbf{u}(t)$. In the mass-spring oscillator example the KF estimates were only forced by $0.5 \cdot \mathbf{q}(t)$, and there was no knowledge of the white noise component $\varepsilon(t)$. This is an example where there exists an unknown forcing that would be distributed by $\Gamma$. The reviewer is correct that here $\mathbf{B} \equiv \Gamma$, but we could have designed the experiment where only mass three (for one example) was forced by a white noise component, which forces $\Gamma$ to be a different operator than $\mathbf{B}$.

*L. 523, "innovation equation": Please identify or report this equation in the manuscript.*

We identify eqn. (4) as the innovation equation.

*L. 551-552, "... depend upon ... and with many observations including those ...": Please rephrase.*

We have changed the sentence as follows:
"The various time-scales embedded in $\mathbf{L}$ depend upon those in $\mathbf{A}, \mathbf{P}(t, -), \mathbf{P}(t)$, and upon the observations entering the KF, including the observational sampling intervals and the structure in the observational noise."

**References**

Bryson, A. E. and Ho, Y.-C.: Applied optimal control, revised printing, Hemisphere, New York, 1975.

Pedlosky, J.: A study of the time dependent ocean circulation, Journal of Atmospheric Sciences, 22, 267–272, 1965.

---

## Author Comment (AC2)

**Response to Reviewer #2**

We thank reviewer #2 for providing thoughtful comments on the manuscript, all of which are greatly appreciated. We first address the major comment suggested, and an enumerated list of responses to the minor comments follows. Reviewer's comments are reproduced in *italic*.

The major comment provided was to summarize and to move much of the content of the introduction to the appendix, and instead to expand on the consequences:

> *The paper addresses an important topic that is often overlooked or ignored. The respective debate goes on for quite some time and is definitely picked up and addressed in the paper by Bengtson et al. The paper is cited, but primarily with respect to observational inhomogeneities. The paper itself makes a very clear statement in the same direction as is being done here: that estimates of the past and future climate needs to be dynamically consistent and preserve basic principles. Might be useful to stress this latter point more beyond the reference to the Dee paper.*
>
> *Having this said, I do like the paper on the one hand because of its message. However, similar to a pre-curser of the paper, it has a very strong text-book or tutorial character and over large parts agrees with the text books of Carl Wunsch. I do understand the rational for including the material. But I also think that the paper can benefit substantially by moving much of this into the appendix and, instead expand on the messages that should be conveyed (see also below). In a nutshell, they could be summarized as:*
>
> *Changes in important functions such as total mass and energy, or volumetric current transports, usually are impacted in non-physical ways by (sequential) data assimilation approached, hindering the process of inferring climate change related trends from reanalyzes. This is because entry of data leads to violation of conservation and other invariant rules. This holds for filter approaches, but even in finite interval smoothing method uncertainties in all such estimation results remain. This message is important. However, I wish a bit more emphasis would have been given on the cure so that the reader is not left with a hopeless feeling.*

We thank the reviewer for their suggestion regarding the manuscript's style and forward, but beg to disagree. The introduction and much of the manuscript is already much shortened compared to the first version that this reviewer saw. The shorter version of the introduction as is provides a useful – and necessary – self-contained formal framing of the problem. Furthermore, changing the format of the manuscript to a Commentary would make it a different manuscript altogether.

We concur with the suggestion to expand more on how to remedy ("cure") some of the underlying issues. In response, we now provide an extensive final paragraph in the Conclusions section. We note that a sequel paper is in preparation, which discusses some of these methods in detail. Adding them here would make this manuscript too long.

We now go over the minor comments provided.

1. *"The papers is written in places in ways that I think are bit cryptic and definitely can be improved. As an example, the first sentence of the abstract is sitting there and the reader wonders what is the connection to the rest of the abstract? Same with the last sentence: here I wish the authors would make a clear statement about what can be done and what cannot be done."*

   We thank the reviewer for these suggestions. In response, we have rewritten the abstract, and have attempted to better clarify the connection from beginning to end. In particular, the first sentence has been altered to read:
   " In sequential estimation methods often used in oceanic and general climate calculations of the state and of forecasts, observations act mathematically and statistically as source or sink terms in conservation equations for heat, salt, mass, and momentum."

   Discussion – even abbreviated – of what can and cannot be done would vastly exceed the space for the abstract, but we have added a brief sentence at the end as follows:
   "Application of smoother-type methods that are designed for optimal reconstruction purposes alleviate some of the issues."

2. *"While re-writing the abstract, I suggest deleting "equation" in line 5 and spell out what "many of the issues" are."*

We clarify what issues we mean by altering the abstract to now read:

"In sequential estimation methods often used in oceanic and general climate calculations of the state and of forecasts, observations act mathematically and statistically as source or sink terms in conservation equations for heat, salt, mass, and momentum. These artificial terms obscure the inference of the system's variability or secular changes. Furthermore, for purposes of calculating changes in important functions of state variables such as total mass and energy, or volumetric current transports, results of both filter and smoother-based estimates are sensitive to mis-representation of a large variety of parameters, including initial conditions, prior uncertainty covariances, and systematic and random errors in observations. Here, toy models of a coupled mass-spring oscillator system and of a barotropic Rossby-wave system are used to demonstrate many of the issues that arise from such mis-representations."

3. *"The first sentence of the introduction is hard to read. I suggest to re-write. Same with the sentence starting line 27. In fact, I suggest re-writing the entire paragraph."*

We have edited the first paragraph in an effort to make it easier to read. Identical steps were taken with the paragraph that began on line 27 (line 30 in the edited manuscript).

4. *"Delete "specifically" on line 29 and identify what you mean by "system trends" on line 31."*

The word "specifically" has been removed, and we have expanded on what is meant by "trend". The final sentence in the paragraph starting around L. 30 now reads:
"In climate science particularly, violations undermine the ability to determine system trends in physical quantities such as temperature, mass, as well as domain-integrated diagnostics (integrated heat and mass content) over months, decades, and longer."

5. *"Line 50: "what is meant by "system failure"? I suggest dropping system here too. Also, this paragraph could use a re-write and definition of what is meant, e.g., by "long-duration forecasts with rigorous models"? Why are rigorous models only "likely to preserve"? Aren't rigorous models defines as preserving quantities (besides numerical effects)?"*

We have now clarified the meaning of "system failure" by the following rewording:
"For some purposes, e.g., short-term weather or other prediction, the failure of the forecasting procedure (consisting of cycles of producing analysis increments from data followed by model forecast) to conserve mass, energy or enstrophy may be of no concern ..."

We also now clarify the difference between a rigorous model and its proper numerical implementation as follows:
"In long-duration forecasts with rigorous models, which by definition contain no observational data at all, conservation laws and other invariants of the model are preserved, provided their numerical implementation is accurate."

6. *"Around line 60 a reference to a textbook defining, e.g., a Kalman filter, would be useful."*

We have now added references to the textbooks by Bryson and Ho (1975) and Wunsch (2006).

7. *"In Section 2, several symbols are either used and only later defined (e.g., $\Delta t$ on line 67). Or they are being shown with the context coming later (the $\tilde{x}(t, -)$ or $\tilde{x}(t, +)$ on line 78)."*

An explanation of $\Delta t$ has now been added. The justification of notation $\tilde{x}(t, -)$ and $\tilde{x}(t, +)$ is properly addressed when these variable are first introduced (now around line 85): "As borrowed from control theory convention, the minus sign in $\tilde{x}(t, -)$ denotes a prediction of $\mathbf{x}(t)$ not using any data at time $t$ ..."

8. *"Line 91 and following: why do you need to go to time step t + delta t before you merge a model and observations? I think you can do the entire argumentation here with x(t,-). If not, I would not understand why."*

The reviewer is correct in noting that the argument could be done with timestep $t$ (instead of $t + \Delta t$), that is, we could assume there exists data $\mathbf{y}(t)$ and we proceed to describe how to compute the weighted average of $\mathbf{y}(t)$ and $\tilde{\mathbf{x}}(t, -)$.

Nevertheless, the discussion would end up identical ours. Our choice to conduct data assimilation at timestep $t + \Delta t$ makes explicit the roles of the model operator $\mathbf{A}$, the known forcing $\mathbf{q}$, and the analysis increment, all within the same equation. The equations as written are self-consistent (e.g., to see this, plug eqn. (4.50) into eqn. (4.52) in Wunsch (2006)).

9. "*Section 3.1.1 and following: I don't see where $I_6$ was introduced and why the equation shown does imply that there is no observational null space. In fact, "null space" was not introduced.*"

   We have now clarified that $I_6$ is the six dimensional identity matrix. With regards to the second half of the comment, if $\mathbf{E} = I_6$ then $\mathbf{E}$ is full rank. Therefore, $\mathbf{E}$ has a vanishing nullspace and we conclude no observational nullspace exists. It is correct that no formal definition of a nullspace has been given in the manuscript, but we opted to omit one as it is an elementary definition from linear algebra.

10. "*With reference to Fig 4 several statements are being made; but I am at lost which panel, which line statements refer to and where I should see what is being said. I suggest to re-write and expand. Holds also for other figures. While you re-write: I have trouble finding line patter (e.g., dahs-dotted) and reding lines.*

    We now refer to specific panels within figures in the text. For example, L. 208 now reads:
    "Until the first data point (denoted with a vertical line in Fig. 4 in all three panels) ..."
    to clarify where we mean the vertical line to exist. We have done our best to distinguish between lines by having both different colors and line patterns. Where these lines coalesce (as they will under some circumstances, e.g., in the initial phase of the KF), these lines will fall on top of each other (e.g, times 0–5000 in Panel b of Fig. 4). Once the their values differ ($t > 5000$) they are well separated and discernable.

11. "*Fig 11 shows in its middle panel a zoom of the top panel. But now you suddenly change line pattern. I suggest keeping them to avoid confusion. Yellow lines are always hard to red. In the bottom panel of Fig 11, I suggest to use two solid lines. You have different colors anyway.*"

    Figure 11 has been recreated so that the patterns in panels (a) and (b) match. Regarding the panel (c) we have opted to stick to solid and dashed (also see previous comment about how to improve distinguishing between lines). This ensures that in the portions where the differences are similar one can see that there are in fact two lines. If both are solid one color overtakes the other and it appears as though there is only one line on the plot.

12. "*In the discussion reference is being made to ECCO, stating that identical results would be obtain in a linear case. This leaves the reader wondering: are all ECCO results equally impacted and corrupted as the once discussed here. The paper would benefit significantly by closing the circle and providing a crips discussion of what the authors belief can be accomplished at all and if in deed al results are the same or where in fact differences exist.*"

    We thank the reviewer for this valid point. We have now slightly restructured the Conclusions section and added an extensive final paragraph that addresses this point.

    > The RTS algorithm is only one choice from several approaches to the finite interval estimation problem. Alternatives include the least-squares/Lagrange multiplier approach of the ECCO project, which in the linear case can be demonstrated to produce identical results. This approach guarantees exact dynamical and kinematic consistency of the state estimate (Stammer et al. (2002), Stammer et al. (2016), Wunsch and Heimbach (2007)), a key requirement when seeking physical understanding of the results. Such consistency is ensured by restricting observation-induced updates to those that are formally independent inputs to the conservation laws, i.e., initial, surface, or – were relevant – lateral boundary conditions. This ensures no artificial source or sink terms in the conservation equations. The ECCO project has conducted this approach with considerable success over the past two decades, and demonstrated the merits of accurate determination of heat, freshwater and momentum budgets and their constituents Heimbach et al. (2019). The Lagrange multiplier framework provides a general inverse modeling framework, which addresses several estimation problems, either separately, or jointly: (i) inference of optimal initial conditions, such as produced by incremental 4-dimensional variational data assimilation practiced in NWP; (ii)

inference of updated (or corrected) boundary conditions, such as practiced in flux inversion methods; (iii) inference of optimal model parameters, such as done in formal parameter calibration problems; or (iv) any combination thereof. A detailed exposure of this more general framework in the context similar toy models will be presented in sequel paper.

**References**

Heimbach, P., Fukumori, I., Hill, C. N., Ponte, R. M., Stammer, D., Wunsch, C., Campin, J.-M., Cornuelle, B., Fenty, I., Forget, G., et al.: Putting it all together: Adding value to the global ocean and climate observing systems with complete self-consistent ocean state and parameter estimates, Frontiers in Marine Science, 6, 55, 2019.

Stammer, D., Wunsch, C., Giering, R., Eckert, C., Heimbach, P., Marotzke, J., Adcroft, A., Hill, C., and Marshall, J.: Global ocean circulation during 1992–1997, estimated from ocean observations and a general circulation model, Journal of Geophysical Research: Oceans, 107, 1–1, 2002.

Stammer, D., Balmaseda, M., Heimbach, P., Köhl, A., and Weaver, A.: Ocean data assimilation in support of climate applications: status and perspectives, Annual review of marine science, 8, 491–518, 2016.

Wunsch, C. and Heimbach, P.: Practical global oceanic state estimation, Physica D: Nonlinear Phenomena, 230, 197–208, 2007.

---

## Referee Report (RR1)

The comments from my review have been addressed to a satisfactory degree, and I think the present manuscript is acceptable for publication. As stated in my previous review, this manuscript should be a significant contribution, as it should invite researchers to interpret cautiously apparent trends or similar features in reanalysis estimates, which have become an important source of information in climate research.

Below is a list of minor points which the authors might want to consider before publication:

l. 27-30: this short paragraph is largely redundant with the last two sentences on l. 68-70 and could be dropped.

l. 68: "… Results and conclusions are discussed …"

l. 89: perhaps replace "deviations" with "uncertainties" or "errors"

l. 90-91: "… knowledge of the initial state elements and resulting uncertainties is required. A bracket …"

l. 108: "… Goodwin Sin (1984) for a fuller discussion". Although …"

l. 113: "… with uncertainty $\mathbf{P}$(t,-) will be …"

l. 141: one could write more generally "… then the elements of $\mathbf{B}_c$ would be zero except in its second row."

l. 142: Replace $\mathbf{A}$ with $\mathbf{A}_c$

l. 143: Replace $\mathbf{A}$ with $\mathbf{A}_c$ and $\mathbf{B}$ with $\mathbf{B}_c$

l. 158: "Equation (8) is discretized at time intervals .. time-step:"

Equation after l. 175: Please number the equation and explain shortly in the text why this equation is an approximation.

l. 185: reference to figure 3a seems to be in order here: "… and is forced by (Fig. 3a) …"

l. 195: "…, as fully unknown, i.e., $\varepsilon(t) \neq 0$."

Table 1 is useful but I think it could be improved. For example, write "observational noise standard deviation is …" (two occurrences), write "$x_0 = (1,0,2,0,0,0)^T$" (add transpose), the expression "$0.5q(t) - \epsilon(t)$" (two occurrences) is ambiguous (e.g., do the authors mean "$0.5q(t) + \epsilon(t)$"?),

and the expression "$x_0 = e_1$" should be replaced with "$x_0 = (1,0,0,0,0,0)^T$ except in section 3.1.2" ($e_1$ is not defined and different initial conditions are assumed in section 3.1.2).

l. 204-205, "Noise with standard deviation 0.01 is added to the observations": This value conflicts with the value of 0.1 in Table 1. Please remove the contradiction. For completeness, please also specify the distribution (normal one?) assumed for the observational noise.

Caption of figure 6: "(a) Correct value ... $x_3(t)$ (blue line) and the estimated value from the KF (red line) ... $x_6(t)$ (blue line) ... from the KF (red line) with error bar ..."

l. 247-248, "... and occur in the two different sets of periodic time intervals": Please specify in the text what these intervals are.

l. 262: "... via the dynamical equations (4). Bias errors ..."

l. 266: Replace "commonly" with "sometimes"?

Fig. 8 is not explicitly referred to in the text. Please refer to this figure in the text, or drop it.

l. 284: ".... $x_5(t)$, its uncertainty ... $P(t)^{-1}$ gives ..."

l. 308-315: Could you provide a numerical illustration, based on one or two of the examples in the manuscript (mass-spring oscillator and Rossby wave model), that the state variables computed from the free-running model equations with the adjusted initial conditions and adjusted control from the RTS are identical (within roundoff errors) to those computed from the RTS equation (19)?

l. 371: "... but model equations with adjusted initial conditions and adjusted parameters and thus ..."

l. 322-323: ".... the variation of the Coriolis parameter, f, with the latitude coordinate, y."

I think equation (22) should be written as

$$\frac{\partial \nabla'^2 \psi}{\partial t'} + \frac{\beta L}{f} \frac{\partial \psi'}{\partial x'} = 0$$

Equation (29): on the right-hand side, the first term represents the contribution from the western boundary layer and the second the contribution from the interior. It is surprising that the interior contribution does not involve beta ... (cf. comment from 1st review).

Equation (30): shouldn't the last element of $\boldsymbol{x}(t)$ be $\psi_s$, not 1?

l. 367: "... given by $\exp(-i\sigma_j \Delta t)$, and ..."

l. 389: Please define $b$.

l. 406: "... Observations cease after $T_f$, mimicking ..."

Figure 10: the stream function minima in the interior suggests the calculations are for a subpolar gyre. This could perhaps be clarified in the text and/or the figure caption.

l. 424: Perhaps extend the last sentence as follows: "... in the interpretation of results and in the development of observing strategies."

l. 431: "..., the control vector of $\boldsymbol{q}(t)$ in this interval, ..."

l. 463-465: "... concern (ECCO ... Lagrange multipliers; Stammer et al. 2002). For ..."

l. 496: "Use of Kalman filters and the simple analogues ... produce ..."

l. 521: "... constituents (Heimbach et al. 2019). The Lagrange ..."

l. 523: "... practiced in numerical weather prediction; (ii) ..."

l. 536: $\boldsymbol{\Gamma}$ is the distribution matrix for the stochastic forcing in the state-transition equation.

l. 473: Should "0" be in bold, e.g., "$\mathbf{G}(t) = \mathbf{0}$"?

l. 581: "... to time $T_f$ so that ..."

---

## Author Response (AR2)

**Response to final review**

We'd again like to thank the reviewer for taking time to go over the manuscript and providing extensive, helpful comments. All of the comments provided in the review have been addressed and applied where necessary. A handful of them are responded to directly below:

**l. 27-30**: We've instead dropped the sentences on l. 68-70, keeping the short paragraph.

**Comments on Table 1**: Transpose indicators have been added, but the definition for forcing $0.5\,\mathbf{q}(t) - \varepsilon(t)$ is exactly what was intended. The forcing function $\mathbf{q}(t)$ is defined by Eq. (17), and we assume no knowledge of the white noise portion giving the $-\varepsilon(t)$. We don't believe this line is ambiguous as there is no other function $\mathbf{q}(t)$ defined. We have also removed the line stating $\mathbf{x}_0 = \mathbf{e}_1$ from the caption.

**l. 204 - 205**: The typo has been corrected, and we have clarified that the observational noise was chosen from a Gaussian distribution in the caption of Table 1.

**l. 247-248**: We have now included the following sentence to clarify the time intervals, "The first interval has observations every 300 timesteps, and the second every 125 timesteps."

**Regarding Figure 8**: We have a reference to Fig. 8 on L. 278.

**l. 308-315**: Due to the length of the manuscript, we prefer not to include further figures.

**Comment on equation 22**: We note that $\beta' = L\beta/f$ when $L = a$, as stated in the comment, but since we're working with the non-dimensional variables we keep just $\beta'$.

**Comment on Eq. (29)**: The approximate solution considered here is that considered in Pedlosky (1965), eqn. (5.11).

**Comment on Eq. (30)**: The last element of the state vector should indeed be 1. The state vector is defined as the time-dependent coefficients of the analytical solution, and since the steady solution $\psi_s$ has no time-dependence the coefficient is just 1.

**l. 367**: We're opting to leave the diag indicator. This emphasizes that we mean a vector of values and not a single exponent for some $j$.

**l. 389**: We've added the sentence "The constant $b$ is chosen as $1.8e-3$ in the numerical code."

**Comment on Fig. 10**: The following sentence has been added to the caption of Fig. 10: "The Coriolis parameter $f$ is computed at a latitude of 30 degrees."